https://doi.org/10.1038/s41467-019-12006-x | **OPEN**

# The molecular basis of chaperone-mediated interleukin 23 assembly control

Susanne Meier[1], Sina Bohnacker[1,2], Carolin J. Klose [1], Abraham Lopez [1,3], Christian A. Choe[4], Philipp W.N. Schmid[1], Nicolas Bloemeke[1], Florian Rührnößl[1], Martin Haslbeck[1], Julia Esser-von Bieren[2], Michael Sattler[1,3], Po-Ssu Huang[4] & Matthias J. Feige[1,5]

The functionality of most secreted proteins depends on their assembly into a defined quaternary structure. Despite this, it remains unclear how cells discriminate unassembled proteins en route to the native state from misfolded ones that need to be degraded. Here we show how chaperones can regulate and control assembly of heterodimeric proteins, using interleukin 23 (IL-23) as a model. We find that the IL-23 α-subunit remains partially unstructured until assembly with its β-subunit occurs and identify a major site of incomplete folding. Incomplete folding is recognized by different chaperones along the secretory pathway, realizing reliable assembly control by sequential checkpoints. Structural optimization of the chaperone recognition site allows it to bypass quality control checkpoints and provides a secretion-competent IL-23α subunit, which can still form functional heterodimeric IL-23. Thus, locally-restricted incomplete folding within single-domain proteins can be used to regulate and control their assembly.

---

[1] Center for Integrated Protein Science Munich (CIPSM) at the Department of Chemistry, Technical University of Munich, Lichtenbergstr. 4, 85748 Garching, Germany. [2] Center of Allergy & Environment (ZAUM), Technical University of Munich and Helmholtz Zentrum München, Biedersteiner Str. 29, 80802 Munich, Germany. [3] Institute of Structural Biology, Helmholtz Center Munich, Ingolstaedter Landstr. 1, 85764 Neuherberg, Germany. [4] Department of Bioengineering, Stanford University, 443 Via Ortega, Stanford, CA 94305, USA. [5] Institute for Advanced Study, Technical University of Munich, Lichtenbergstr. 2a, 85748 Garching, Germany. Correspondence and requests for materials should be addressed to M.J.F. (email: matthias.feige@tum.de)

In order to become functional, a large number of proteins depend on assembly into higher order complexes[1–3]. Assembly thus needs to be aided and scrutinized by molecular chaperones that surveil the multiple steps of protein biosynthesis from translation on the ribosome to adopting the final native structure[4]. In fact, unassembled proteins likely represent a major class of clients for the cellular quality control machinery[5,6] but also a particularly complicated one to assess: on the path from protein folding to assembly, the degree of structure in an immature protein can be expected to increase, as specific protein-protein interactions depend on specific interfaces. This simple notion, however, poses a conundrum: chaperones recognize non-native states of proteins and can target their clients for degradation if folding does not occur. Unassembled subunits, on the other hand, need to be stable and structured enough to allow for specific interactions, avoiding futile steps in the biosynthesis of proteins, but also to allow the cellular quality control machinery to read their assembly state. Although specific assembly chaperones exist for particularly abundant and complex clients[7,8], most proteins can be expected to rely on the more generic chaperone machineries to survey their oligomerization state. In the cytoplasm, recent studies have shown that protein translation and assembly can be intimately coupled, increasing efficiency of these processes by spatial constraints[9,10] or translational pausing[11]. Such a scenario has not been described for secretory pathway proteins, which are produced in the endoplasmic reticulum (ER) and make up ca. 1/3 of all proteins produced in a typical mammalian cell[12]. For these, translation in the cytoplasm and assembly in the ER are spatially separated by the translocon. Cells still have to ensure that proteins correctly assemble before being transported to their final destination from the ER, at the same time avoiding premature degradation[13]. Furthermore, as opposed to the cytosol, quality control proteases or ubiquitin conjugating systems are absent from the lumen of the ER, rendering assembly control highly dependent on recognition by the generic ER chaperone machinery[5,14].

In order to better understand the regulation and control of protein assembly processes in its biologically relevant cellular context[15], we thus need to refine our understanding of what chaperones recognize as signatures of unassembled proteins. Although structural insights into chaperone-client interactions exist in some cases[16–22], these remain limited and are mostly absent in vivo. During this study we thus selected a protein model system where assembly control is particularly relevant to maintain proper functioning of the immune system, the heterodimeric interleukin-23 (IL-23)[23]. IL-23 is a key cytokine involved in inflammatory diseases as well as cancer and has become a major therapeutic target in the clinics[24–27]. It is composed of one α-and one β-subunit, which need to assemble in order for the cytokine to be secreted[23]. We show that locally restricted incomplete folding of one subunit allows for reliable assembly control of the heterodimeric protein by ER chaperones while at the same time avoiding premature degradation of unassembled subunits. Structural insights into IL-23 biogenesis and chaperone recognition allow us to rationally engineer protein variants that can pass quality control checkpoints even while unassembled. Engineering such variants may provide proteins with new biological functions in cellular signaling and immune regulation.

## Results

**Assembly-induced folding regulates IL-23 formation.** IL-23 is a heterodimeric cytokine composed of IL-23α and IL-12β (Fig. 1a). IL-23α alone is efficiently retained in cells and IL-12β induces its secretion[23] (Fig. 1b) as one well-defined, covalent IL-23α/IL-12β heterodimer[23,28] (Fig. 1c). In contrast, unassembled, intracellular IL-23α showed multiple disulfide-bonded species on non-reducing SDS-PAGE gels (Fig. 1c). Thus, IL-23α fails to fold into one defined native state in the absence of IL-12β and (some of) its cysteines remain accessible while unpaired with IL-12β. A closer scrutiny of the IL-23α structure revealed three different types of cysteines within the protein: (1) C58 and C70, which form the single internal disulfide bond (2) C54, which engages with IL-12β upon complex formation, stabilizing the IL-23 heterodimer by a disulfide bond[23,28] and (3) two free cysteines (C14, C22) in the first helix of its four-helix bundle fold (Fig. 1d). Cysteines are among the evolutionary most highly conserved amino acids and the presence of free cysteines in secretory pathway proteins is rare, as they may induce misfolding and are often recognized by the ER quality control (ERQC) system[29]. This suggests, that the unpaired C14 and C22 of IL-23α may play a functional role.

Structure and sequence alignments of the IL-23α subunit with its homologs IL-6 and IL-12α revealed interesting features about the cysteine configurations in these three proteins: although the single internal disulfide bond is structurally conserved (Fig. 1e and Supplementary Fig. 1a), the disulfide bonding pattern is quite different. In IL-23α, the disulfide bond is formed by residues in sequence proximity, C58 and C70, similar to IL-6 (Fig. 1 e, f). In contrast, in IL-12α, residues more distant in the sequence (C63 and C101) form the corresponding disulfide bond. This leaves C80 in IL-12α free to engage with C15 in its first α-helix, forming a second disulfide bond (Fig. 1e, f). This is unlike IL-23α, where C14 and C22 in its first α-helix remain unpaired but are buried in the native structure of IL-23α in the IL-23 heterodimer (Supplementary Fig. 1b).

Based on these observations we assessed the function of the different types of cysteines in IL-23α for its assembly-induced secretion by IL-12β. A very small amount of unpaired FLAG-tagged IL-23α was secreted for wild-type IL-23α (IL-23α$^{wt}$) and similarly for mutants of the different types of cysteines (Fig. 1b, g), but the effect of mutating cysteine residues in IL-23α on assembly-induced secretion was very pronounced: Replacing the disulfide bond-forming cysteines (IL-23α$^{C58,70S}$) led to a drastically reduced effect of IL-12β on promoting IL-23α secretion, indicating a key structural role for these cysteines for assembly-induced secretion (Fig. 1g). Replacement of C54 by Ser had an intermediate effect on assembly-induced secretion by IL-12β (Fig. 1g). Surprisingly, replacement of the two free cysteines in helix 1 of IL-23α by valines (IL-23α$^{C14,22V}$) or serines (IL-23α$^{C14,22S}$) did not have any pronounced effect on the efficiency of IL-12β-induced secretion of IL-23α (Fig. 1g and Supplementary Fig. 1c). The fact that the two unusual free cysteines in IL-23α are dispensable for efficient formation of the IL-23 heterodimer suggests that they fulfil other roles, which led us to characterize their molecular functions in more detail.

**Cysteines in IL-23α act as stabilizing chaperone anchors.** We first assessed the influence of the different types of cysteine residues in IL-23α on its stability in cells. IL-23α$^{wt}$ was degraded with a half-life of $63 \pm 11$ min (Fig. 2a). In agreement with a key structural role, deleting the disulfide bond-forming cysteines in IL-23α$^{C58,70S}$ led to a faster degradation with a half-life of $30 \pm 9$ min (Fig. 2a). Quite unexpectedly, however, the IL-23α$^{C14,22V}$ and IL-23α$^{C54S}$ mutants were degraded even faster with half-lives of only $20 \pm 7$ min and $17 \pm 5$ min, respectively (Fig. 2a). A similar half-life was observed for IL-23α$^{C14,22S}$ (Supplementary Fig. 2a). Accordingly, the unpaired cysteines in helix 1 of IL-23α do not significantly promote IL-23 heterodimerization—but stabilize the protein while unassembled. This behavior can be explained by two hypotheses: either these cysteines affect IL-23α misfolding in

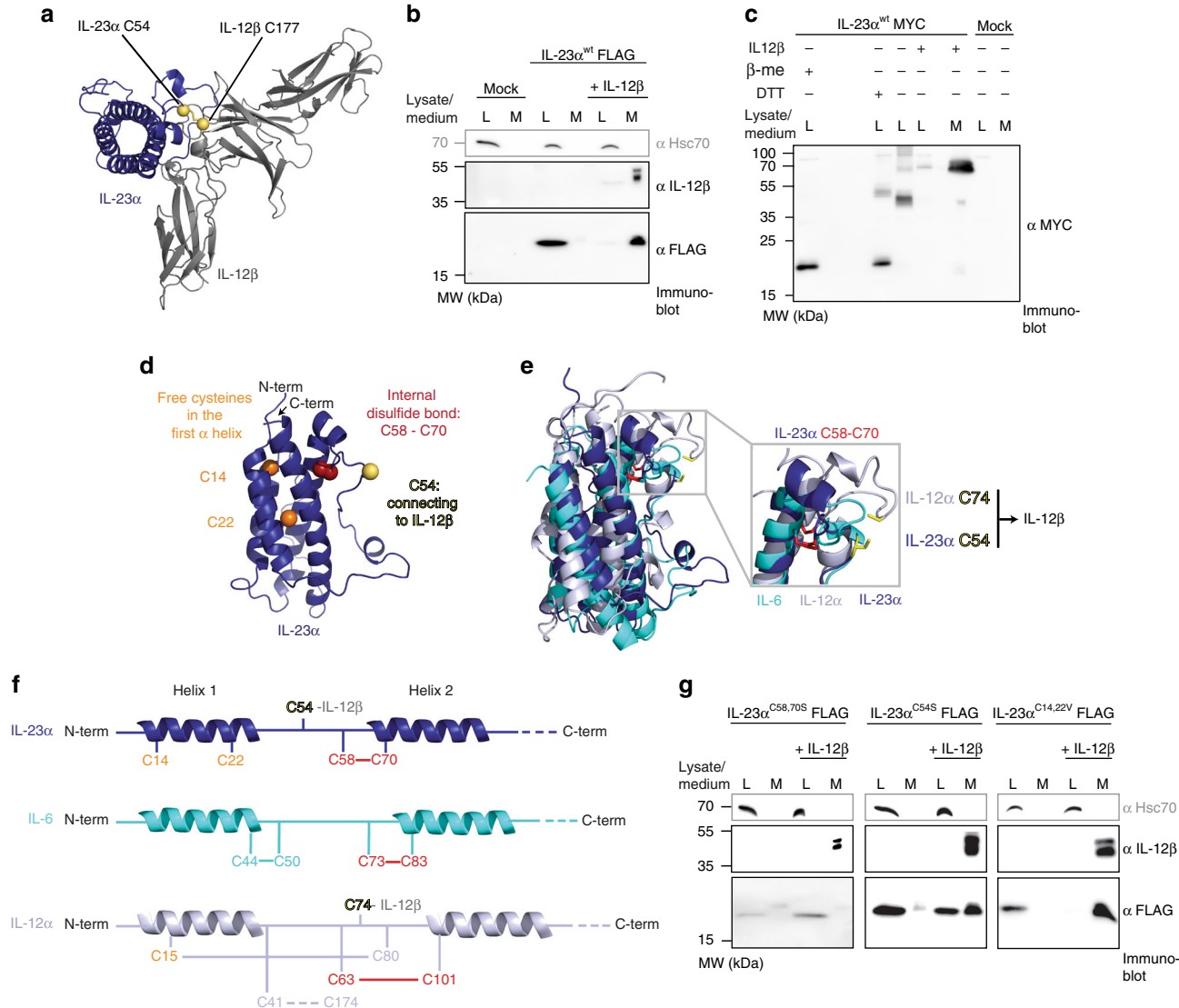

**Fig. 1** IL-23α misfolds in cells in the absence of IL-12β. **a** Structure of heterodimeric IL-23. Cysteines in IL-23α (blue) and IL-12β (gray) that form an intermolecular disulfide bond are shown in yellow. **b** Secretion behavior of FLAG-tagged wild type IL-23α (IL-23α[wt]) in the presence or absence of its interaction partner IL-12β. Hsc70 served as loading control. **c** IL-23α forms non-native disulfide bonds in isolation (lane 3) and IL-12β covalently heterodimerizes with IL-23α (lanes 4 and 5), concomitantly reducing misfolding of IL-23α. Samples were treated with β-Me post-lysis/DTT in cells for reduction where indicated and with NEM to conserve redox species. **d** Structure of IL-23α. Cysteines that form an intramolecular disulfide bond in IL-23α are shown in red, the one that engages with IL-12β is highlighted in yellow, and free cysteines are depicted in orange. **e** Structural alignment of IL-23α (blue), IL-6 (cyan) and IL-12α (light gray). The conserved disulfide bond is shown in red and the IL-12β engaging free cysteines of IL-23α and IL-12α in yellow. **f** Model of IL-23α, IL-6 and IL-12α illustrating cysteines and disulfide bonds. The same color code as in **d, e** was used. Numbering is without signal sequences. **g** Secretion behavior of FLAG-tagged IL-23α constructs as in **b** but with the indicated IL-23α cysteine mutants

isolation and/or they are recognized differently by the ER quality control system. The latter could provide valuable insights into how protein folding states are recognized on a molecular level in the ER.

All IL-23α mutants that still contained free cysteines showed a similar degree of misfolding and misassembly (Supplementary Fig. 2b, c). We thus proceeded to test the second hypothesis, that the cysteines are recognized differently by chaperones. Unpaired cysteines in secretory pathway proteins can be recognized by protein disulfide isomerase (PDI) family members in the ER[30]. Since we did not observe any significant difference in binding of PDI itself to IL-23α[wt] versus IL-23α cysteine mutants (Supplementary Fig. 2d), we assessed interaction with another PDI family member, ERp44. ERp44 serves as an ER recruitment chaperone from the ER–Golgi intermediate compartment (ERGIC) during

protein assembly[31] and thus was an interesting candidate in terms of IL-23 assembly control. IL-23α[wt] strongly bound to ERp44 (Fig. 2b) and was partially co-localized with ERp44 in the ERGIC (Supplementary Fig. 2e) indicating a biologically relevant interaction. This was further confirmed by a transient knock-down of ERp44, which led to partial secretion of unassembled IL-23α (Supplementary Fig. 2f). Of note, binding of ERp44 was significantly reduced for IL-23α[C14,22V] and IL-23α[C54S] versus IL-23α[wt], whereas binding to the IL-23α[C58,70S] mutant was not affected (Fig. 2b). Single cysteine mutants in helix 1 of IL-23α (IL-23α[C14S] and IL-23α[C22S]) also showed reduced binding to ERp44, which was significant for the C14S mutant (Supplementary Fig. 2g).

To additionally assess if any chaperones act upstream of ERp44 on IL-23α, i.e.: in the ER, we analyzed binding of the ER Hsp70

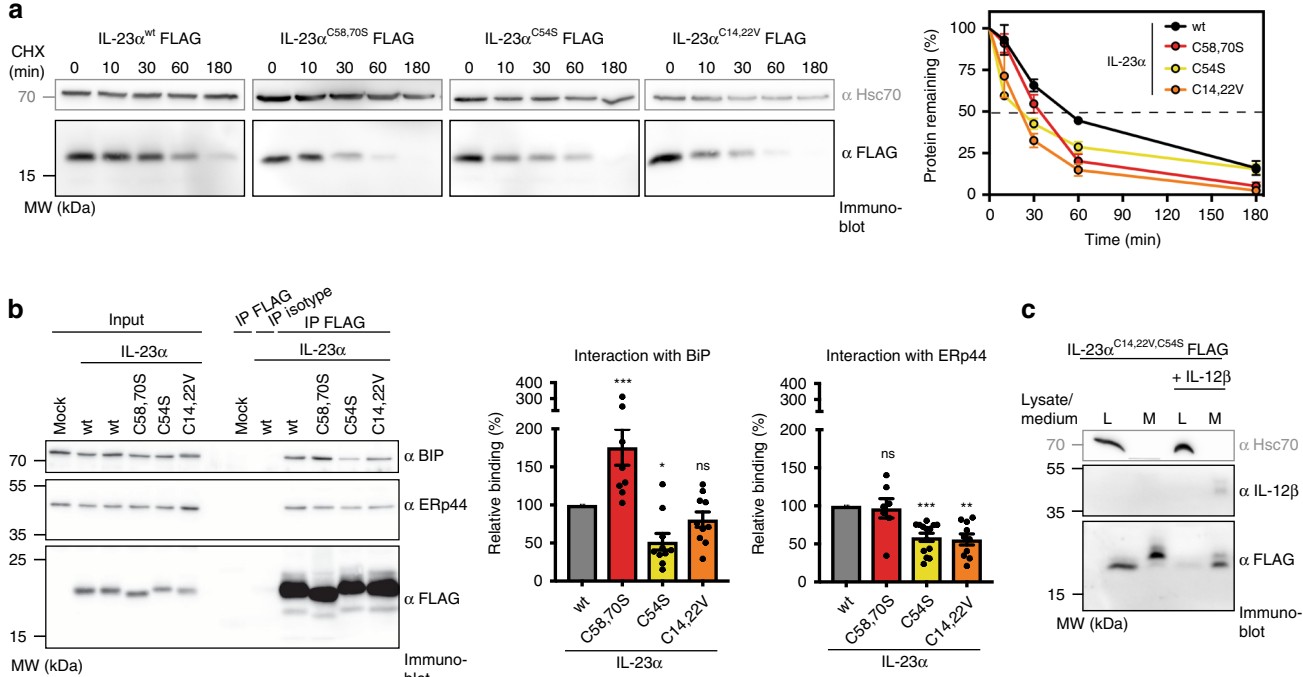

**Fig. 2** Cysteines in IL-23α are recognized by ER chaperones and influence its half-life. **a** Time course of degradation for IL-23α$^{wt}$, IL-23α$^{C58,70S}$, IL-23α$^{C54S}$, and IL-23α$^{C14,22V}$ upon inhibition of translation with cycloheximide (CHX) for up to 180 min. Hsc70 served as loading control. Right: Quantification of protein turnover is shown; data are the mean ±SEM of at least four independent experiments. Source data are provided as a Source Data File. **b** Immunoblot analysis of co-immunoprecipitated co-transfected hamster BiP or endogenous ERp44 with FLAG-tagged IL-23α$^{wt}$, IL-23α$^{C58,70S}$, IL-23α$^{C54S}$, and IL-23α$^{C14,22V}$. Center and right: Relative binding was determined from at least seven independent experiments (shown ±SEM) and normalized to the IL-23α$^{wt}$ signal, which was set to 100% relative binding. Statistical significance was calculated using a two-tailed unpaired *t*-test. ns = not significant; *$p < 0.05$, **$p < 0.01$, ***$p < 0.001$ indicate statistically significant differences. The IL-23α$^{C58,70S}$ construct has a shorter linker between the protein and tag, which explains its slightly faster migration. **c** Secretion behavior of FLAG-tagged IL-23α$^{C14,22V,C54S}$ in the presence or absence of IL-12β. Hsc70 served as loading control

chaperone BiP to IL-23α and its cysteine mutants. BiP binds to exposed hydrophobic stretches in proteins and can thus serve as a good proxy to assess the folding status of a protein in the ER[32]. IL-23α$^{C58,70S}$ bound significantly stronger to BiP than IL-23α$^{wt}$ (Fig. 2b), corroborating a stabilizing structural role for the conserved disulfide bond in IL-23α. Mutating cysteine 54 reduced BiP binding, whereas mutating cysteines 14 and/or 22 led to unaltered BiP binding in comparison to IL-23α$^{wt}$ (Fig. 2b and Supplementary Fig. 2g). This is in agreement with the notion that replacing the free cysteines in IL-23α does not disrupt its structure formation.

Taken together, our data suggest that free cysteines in IL-23α, C14, C22, and C54, are bound by ERp44 and increase the half-life of the protein. IL-23α$^{C58,70S}$, in contrast, loses structural stability provided by the internal disulfide bond-forming cysteines, is degraded rapidly and is compromised in its assembly-competency. If this hypothesis was correct, replacing all free cysteines in IL-23α should, in principle, lead to its secretion even while unpaired. Indeed, an IL-23α mutant where C14 and C22 were replaced by Val and C54 by Ser (IL-23α$^{C14,22V,C54S}$) was secreted even in the absence of IL-12β (Fig. 2c). The fact that a significant amount of IL-23α$^{C14,22V,C54S}$ was still retained in the cell, however, indicated the existence of further quality control mechanisms that act on IL-23α maturation and heterodimerization.

**A localized folding transition in IL-23α upon assembly.** Our data show that all unpaired Cys in IL-23α function as a signal for its assembly status. Cysteines 14 and 22, however, are buried in the native structure of IL-23 (Fig. 1d and Supplementary Fig. 1b) pointing towards structural changes occurring in IL-23α upon

assembly. We reasoned that these changes might underlie a second quality control mechanism and could provide rare structural insights into protein assembly control in the ER. To analyze these further and complement our cellular studies by structural analyses of IL-23 assembly control, we decided to study IL-23α in vitro. Based on our cellular studies we proceeded with a mutant where all free cysteines were replaced and only the internal disulfide bond was retained, in order to understand the presumed second QC mechanism which acts independently of free cysteines. Consistent with our cellular studies, C14 and C22 were replaced by valines and the IL-12β-connecting C54 by serine (IL-23α$^{C14,22V,C54S}$, denoted as IL-23α$^{VVS}$ for simplicity in the following). As opposed to the wt protein, which was highly prone to misfolding, IL-23α$^{VVS}$ could be purified to homogeneity and formed its disulfide bond (Supplementary Fig. 3a). Analytical ultracentrifugation (auc) experiments revealed a well-defined monomeric state of IL-23α$^{VVS}$ with a slightly elongated shape as expected from the crystal structure of IL-23 (Supplementary Fig. 3b)[28]. Far-UV CD spectroscopy showed that IL-23α$^{VVS}$ possesses α-helical structure (Fig. 3a), although the overall intensity was low, which argues for flexible regions in the protein. Consistent with this notion, temperature-induced unfolding (Fig. 3b) and partial proteolysis experiments (Supplementary Fig. 3c) revealed an overall rather low stability for IL-23α$^{VVS}$ with a melting point of 48 ± 0.2 °C and a half-life against proteolysis of 18 ± 2 min, respectively. A relatively small spectral dispersion observed in NMR $^1$H, $^{15}$N heteronuclear single quantum coherence (HSQC) experiments was consistent with the notion that IL-23α$^{VVS}$ contains mainly helical and flexible/disordered regions (Fig. 3c).

To further understand the structural features of unpaired IL-23α and possible changes upon formation of the native

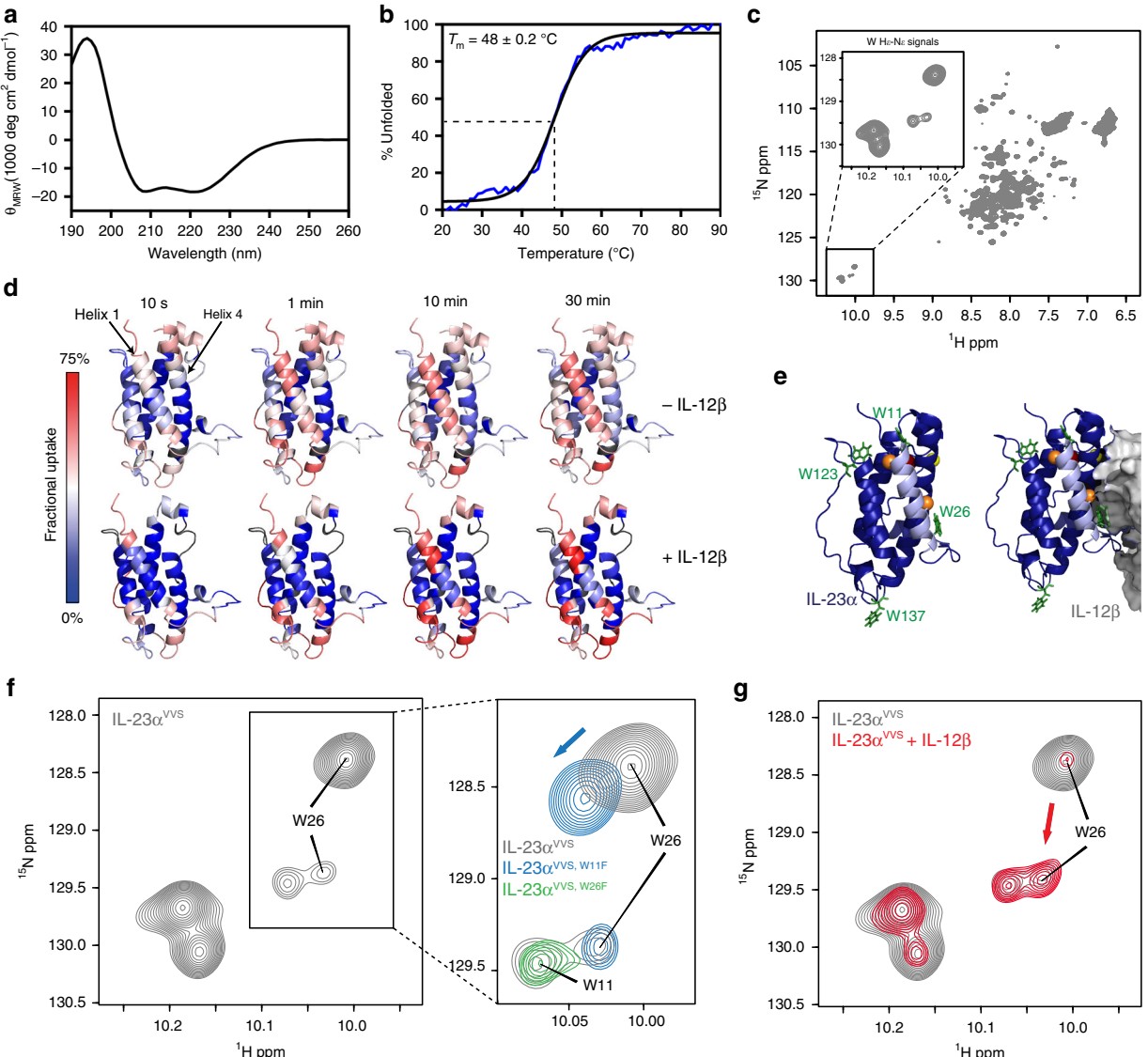

**Fig. 3** Helix 1 in IL-23α gains structure upon assembly with IL-12β. **a** Far-UV CD spectrum of IL-23α$^{VVS}$. **b** IL-23α$^{VVS}$ unfolds cooperatively with a melting temperature of 48 ± 0.2 °C. **c** $^1$H, $^{15}$N HSQC NMR spectrum of IL-23α$^{VVS}$; the inset shows the downfield-shifted tryptophan indole side chain signals as zoomed view. **d** Hydrogen/deuterium exchange (HDX) experiments reveal helix 1 in IL-23α$^{VVS}$ to be highly flexible and to be stabilized upon interaction with IL-12β. IL-23α is colored according to the measured HDX rates. Blue colors correspond to a lower (less flexible regions) and red colors to a higher (flexible regions) fractional uptake (gray: no sequence coverage in HDX measurements). **e** Structure of IL-23α (blue) with helix 1 in light blue and cysteine residues shown, using the same color code as in Fig. 1d and in complex with IL-12β (gray). Trp residues are shown in green. **f** Trp indole side chain signals in $^1$H, $^{15}$N HSQC experiments for IL-23α$^{VVS}$. Unambiguous assignment of W26 from the two minor signals was obtained by analyzing the spectra of IL-23α$^{VVS, W26F}$ (green, zoomed view) and an additional IL-23α$^{VVS,W11F}$ mutant (blue, zoomed view). The intensity of the spectrum for IL-23α$^{VVS, W26F}$ was lower and thus increased two-fold to allow for a comparison. **g** Same as **f** but for unpaired IL-23α$^{VVS}$ (black) versus IL-23α$^{VVS}$ in the presence of a two-fold molar excess of unlabeled IL-12β (red). The intensity of the spectrum for IL-23α bound to IL-12β was increased to compensate the gain in molecular weight of the complex. The same experimental parameters were used for both measurements

heterodimer, we performed hydrogen/deuterium exchange (HDX) measurements on IL-23α$^{VVS}$ and on the IL-23 heterodimer. In the IL-23 heterodimer, C14 and C22 of IL-23α were also replaced by valines, but C54 was preserved to enable the formation of the intermolecular disulfide bond between the IL-23 subunits. HDX measurements revealed an overall higher flexibility for IL-23α$^{VVS}$ in isolation in comparison to the corresponding heterodimer (Fig. 3d and Supplementary Fig. 4). Helix 4 in IL-23α$^{VVS}$, where the major interaction site with IL-12β is located[28], was already relatively stable even when IL-23α$^{VVS}$ was unpaired and was further stabilized upon heterodimerization (Fig. 3d). Of note, the first helix of isolated

IL-23α$^{VVS}$ was the most flexible region within the isolated subunit and became strongly stabilized upon interaction with IL-12β (Fig. 3d). This first helix is exactly the region where the two free cysteines (C14, C22) are located, which we identified to be recognized by ERp44. A similar behavior was observed for another mutant, where the two free cysteines in helix 1 were replaced by serines instead of valines as well as for the wt IL-23 complex (Supplementary Fig. 3d–f and Supplementary Fig. 4), suggesting that this behavior was intrinsic to IL-23α. When complexed with IL-12β, the different IL-23α mutants behaved like the wt protein in a receptor activation assay testing for biological activity (Supplementary Fig. 5). Thus, the structural

changes we observed were fully consistent with formation of functional IL-23.

To further understand IL-12β-induced conformational rearrangements in IL-23α we used NMR spectroscopy. Strikingly, we observed five signals corresponding to tryptophan side chain indole NH groups in the $^1$H, $^{15}$N HSQC spectrum (Fig. 3c, inset), although IL-23α only contains four tryptophans (Fig. 3e). This argues for conformational heterogeneity and dynamics in IL-23α$^{VVS}$ on the time scale of milliseconds or slower, indicating conformations with distinct chemical environments. In order to investigate this further, we assigned those resonances by single-point mutagenesis of individual tryptophan residues. This approach revealed that Trp26 gives rise to two signals in the NMR spectrum (Fig. 3f). Of note, Trp26 is located at the end of helix 1 of IL-23α and in the IL-12β binding interface (Fig. 3e). Thus, our NMR measurements also suggest that helix 1 is conformationally heterogenous, populating two states that are in slow exchange at the NMR time scale. One of these likely corresponds to an incompletely folded form, as indicated by the HDX measurements. If indeed a folding transition involving helix 1 played a role in IL-23 assembly control, as suggested by our cellular data and HDX measurements, this conformational transition should be detectable by NMR. In agreement with this idea, the presence of IL-12β caused the intensity of the major Trp26 indole signal to almost entirely shift towards the pre-existing minor conformation (Fig. 3g). This corroborates that IL-12β induces folding of IL-23α, involving helix 1, and supports the notion that its first α helix is mostly unfolded in the absence of IL-12β.

Taken together, our comprehensive analysis reveals an assembly-induced folding mechanism where IL-12β recognizes structured regions within IL-23α and induces further folding of the entire α-subunit, in particular its first α helix. This reveals important information about what ER chaperones can recognize as signatures of an unassembled protein.

**Structurally optimized IL-23α can bypass ER quality control.** Our analyses revealed the first α helix in IL-23α to be unstructured while this subunit is unpaired, and to gain structure upon heterodimerization with IL-12β. Consequently, the two free cysteines that can otherwise be recognized by PDI chaperones become buried, pointing toward an intricate quality control mechanism that oversees IL-23 assembly. Building on these insights, we wondered if IL-23α could bypass ER quality control by selectively improving the stability of its first helix. Towards this end we optimized helix 1 of IL-23α in silico using RosettaRemodel[33]. The native structure of IL-23 contains a number of non-ideal structural features[34]. Upon first inspection, we found that a few of the residues near the N-terminus can be improved from their native environment (see methods for details). For example, Pro9 is exposed with little structural support; Ser18 is completely buried, and likely interacts with its own helical backbone, which may reduce the rigidity of the structure. We thus redesigned all of the core-facing residues on helix 1, adjusted the buried polar residues to hydrophobic ones, extended the N-terminus of the crystal structure by two residues, and completely rebuilt the first six amino acids in order to create a stable N-terminus. Taken together, this led to three optimized models for IL-23α (Supplementary Fig. 6a), out of which we proceeded with one for experimental testing that had one of the cysteines (C22) in helix 1 still in place (Fig. 4a). This engineered protein is referred to as IL-23α$^{opt}$ in the following. Strikingly, IL-23α$^{opt}$ was independently secreted from mammalian cells (Fig. 4b), despite the presence of C22 in helix 1 of IL-23α$^{opt}$ (Fig. 4a) and the presence of the unpaired C54 residue. Thus, optimization of

the first helix in IL-23α makes IL-12β dispensable for its secretion. Of note, IL-23α$^{opt}$ secreted in absence of IL-12β showed a slightly higher molecular weight than the non-secreted protein (Fig. 4b), which we had observed also for IL-23α$^{VVS}$ (Fig. 2d). We could attribute this increase in molecular weight to O-glycosylation of IL-23α$^{opt}$ occurring at residue T167 (Supplementary Fig. 6b, c). O-glycosylation occurs in the Golgi, and hence IL-23α$^{opt}$ correctly traverses the secretory pathway, indicating proper folding. Apparently, interaction with IL-12β normally blocks this O-glycosylation site, which is consistent with the location of residue T167 in the IL-23α/IL-12β interface (Supplementary Fig. 6c). Indeed, when IL-23α$^{opt}$ was secreted in the presence of IL-12β, it did not become modified (Fig. 4b), again in agreement with data for IL-23α$^{VVS}$ (Fig. 2d). This also shows that, although IL-23α$^{opt}$ can be independently secreted, it is still able to very efficiently assemble into the heterodimeric IL-23 complex (Fig. 4b and Supplementary Fig. 6d), further confirming correct folding of the optimized mutant. If indeed helix 1 was a major chaperone recognition site in the ER, IL-23α$^{opt}$ should not only be secreted but also show reduced chaperone binding. In complete agreement with this hypothesis, chaperone interaction experiments revealed significantly less binding of BiP and ERp44 to IL-23α$^{opt}$ (Fig. 4c).

To structurally characterize IL-23α$^{opt}$ in vitro and validate our engineering efforts, we created an IL-23α$^{opt}$ mutant where the IL-12β-connecting C54 residue was additionally replaced by serine to minimize the aggregation potential of the protein (IL-23α$^{opt, C54S}$). The purified protein formed its internal disulfide bond (Supplementary Fig. 7a) and showed an α-helical conformation (Fig. 4d). Analytical ultracentrifugation experiments revealed the same slightly elongated shape for IL-23α$^{opt, C54S}$ as for IL-23α$^{VVS}$ and a predominantly monomeric state (Supplementary Fig. 7b). IL-23α$^{opt, C54S}$ had a melting point of 61 ± 0.7 °C (Fig. 4e), more than 10 °C higher than observed for the non-optimized protein (Fig. 3b). Unfolding, however, was much less cooperative, which may be due to the strong stabilization of helix 1. Very similar melting points and (non-)cooperativity were observed in near-UV CD thermal transitions, which monitor loss in tertiary structure (Supplementary Fig. 7c, d). Taken together, these data suggest that selective stabilization of helix 1 indeed increases folding efficiency of this structural element (and of the entire protein). Apparently this comes at the cost of losing folding cooperativity, which had been observed previously in a computationally designed protein[35]. Consistent with this notion, HDX measurements revealed the unpaired IL-23α$^{opt, C54S}$ to be overall more stable, including the first optimized helix which was very labile in the non-optimized protein. Furthermore, helix 1 in IL-23α$^{opt}$ was only slightly further stabilized upon complex formation with IL-12β (Fig. 4f and Supplementary Fig. 8). Again, in the complex C54 in IL-23α$^{opt}$ was preserved, to allow for covalent complex formation. Importantly, the IL-23α$^{opt}$/IL-12β heterodimer still showed biological activity, corroborating that IL-23α$^{opt}$ was not only correctly folded but also functional in the context of IL-23 (Supplementary Fig. 5). Taken together we identified the first helix of IL-23α as a major limiting factor in its folding process. Stabilization of this small region stabilizes the whole protein, enabling its escape from ER retention while preserving assembly-competency with IL-12β and biological activity.

## Discussion

In this study we provide structural insights into how ER chaperones can recognize unassembled proteins and aid their assembly into protein complexes—while at the same time preventing their premature degradation. We identified two structural

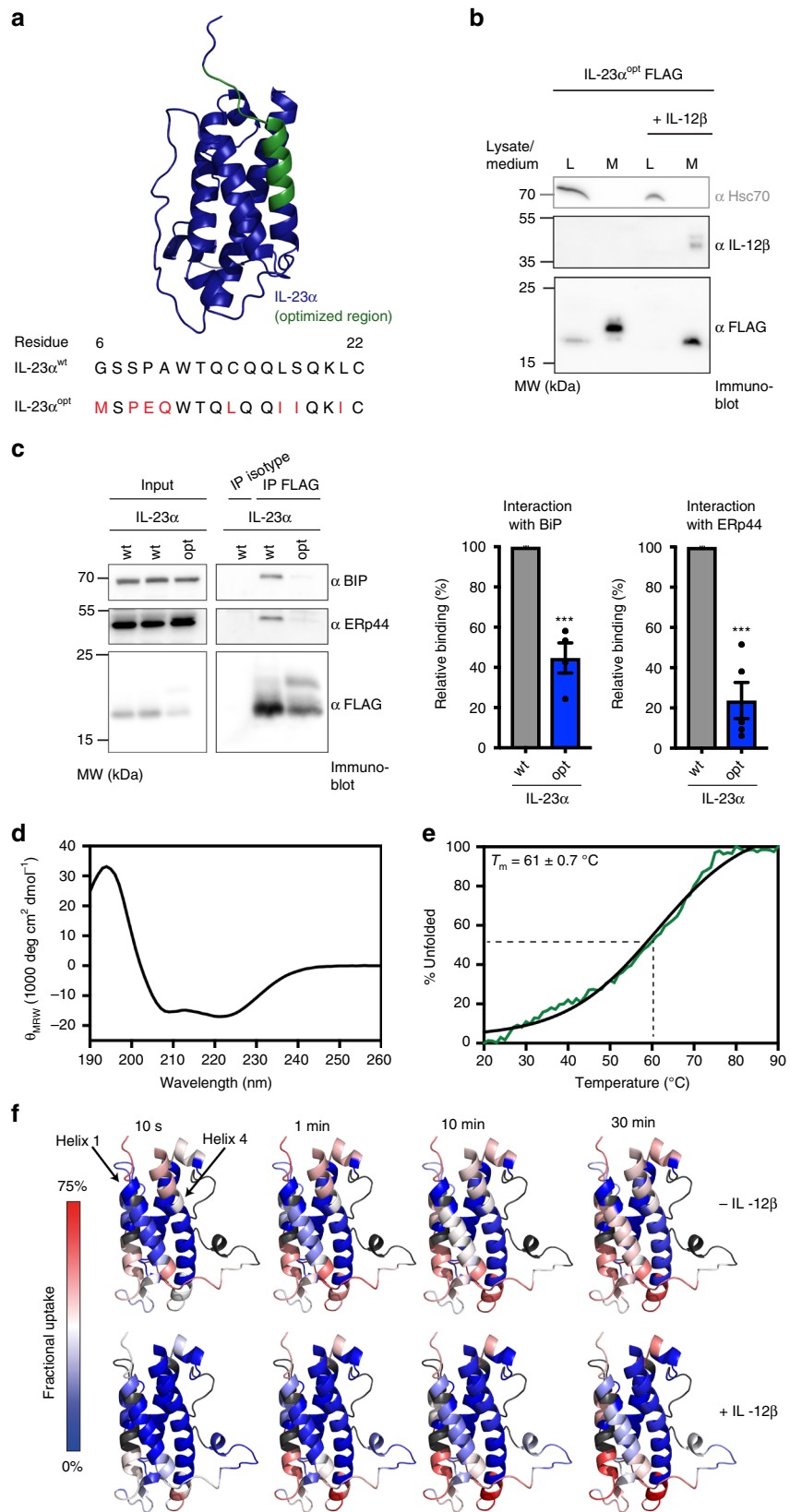

features in IL-23α that synergistically guarantee proper ER quality control and assembly of the potent immune activator IL-23 (Fig. 5): (1) incomplete folding, in particular of its first α-helix, detected by BiP and (2) free cysteines recognized by the PDI family member ERp44. Intriguingly, these two motifs are located in the same region within IL-23α, but would be recognized at different stages of the secretory pathway. BiP is able to recognize hydrophobic stretches in partially unfolded proteins already as early as during co-translational import into the ER[36–38], whereas ERp44 acts later inside the ER–Golgi intermediate compartment[39], preventing secretion of unassembled or incorrectly folded proteins[31]. Our structural analyses combined with cellular studies

**Fig. 4** Optimization of helix 1 allows IL-23α to pass ER quality control in isolation. **a** IL-23α helix 1 optimization. Top: Structure of IL-23α with the optimized region highlighted in green. Bottom: Sequence comparison of amino acids 6–22 of IL-23α[wt] and IL-23α[opt]. Amino acid exchanges in IL-23α[opt] are highlighted in red. **b** Secretion behavior of FLAG-tagged IL-23α[opt] in the presence and absence of IL-12β. Hsc70 served as a loading control. **c** Immunoblot analysis of co-immunoprecipitated co-transfected hamster BiP or endogenous ERp44 with FLAG-tagged IL-23α[opt]. Center and right: Relative intensity of each band was calculated for at least four independent experiments (shown ±SEM) and normalized to the IL-23α[wt] signal which was set to 100%. Statistical significance was calculated using a two-tailed unpaired $t$-test. ***$p < 0.001$ indicates statistically significant differences. **d** Far-UV CD spectrum of IL-23α[opt]. **e** IL-23α[opt] unfolds with a melting temperature of 61 ± 0.7 °C. **f** Hydrogen/deuterium exchange (HDX) experiments of unpaired IL-23α[opt] versus the IL-12β-paired IL-23[opt]. IL-23α[opt] is colored according to the measured HDX rates. Blue colors correspond to a lower (less flexible regions) and red colors to a higher (flexible regions) fractional uptake (gray: no sequence coverage in HDX measurements)

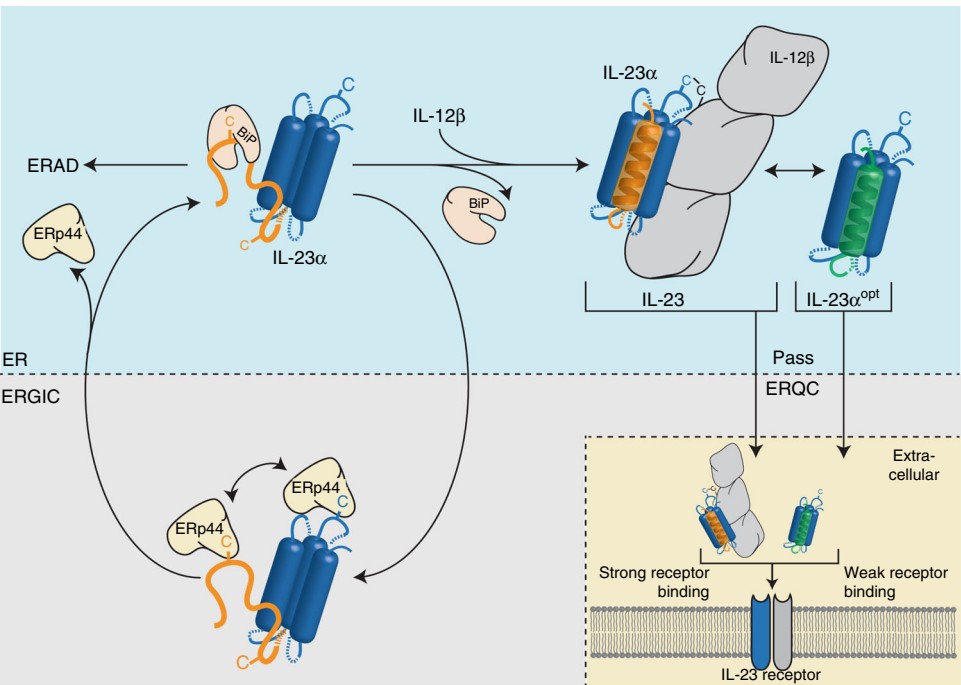

**Fig. 5** A model for IL-23 assembly control in the cell. Incomplete folding of IL-23α[wt] is recognized by chaperones along the secretory pathway. IL-23α is incompletely structured in isolation, in particular the first out of its four helices, and can be recognized by BiP during early biogenesis steps within the ER. ERp44, a member of the PDI-family, supports BiP function by retrieving IL-23α from the ERGIC compartment to the ER, thus acting downstream of BiP. BiP and ERp44 act together, to maintain assembly competency of IL-23α. Upon assembly with IL-12β, IL-23α completes folding of its first helix, which inhibits chaperone interaction and results in secretion of the heterodimeric IL-23 complex, connected by a disulfide bond. If IL-23α does not assemble properly, it is targeted for ER-associated degradation (ERAD). ERAD is slowed down by the presence of free cysteines in IL-23α, thus most likely by chaperone binding. Stabilization of the first helix renders IL-23α insensitive to chaperone interaction and allows independent folding and secretion. Despite independent secretion, IL-23α[opt] is still able to interact with IL-12β. IL-23 induces strong signaling upon receptor binding, whereas IL-23α[opt] shows weak receptor activation. Loops within the structure of IL-23α are indicated as dashed lines

thus allow us to understand, how ER protein assembly can be controlled with high fidelity by sequential quality control checkpoints, which is conceptually reminiscent although distinct on a molecular level to IgM antibody assembly control[17,40–42]. It remains to be seen, if a competition for BiP and ERp44 exists for binding to IL-23α and if binding differences would entail different fates. Furthermore, our study provides insights into how premature degradation of unassembled proteins may be avoided: The first α-helix of IL-23α, which we identified to be an incompletely folded chaperone recognition site, is devoid of any sequence patterns that would allow binding to ERdj4, ERdj5 or Grp170 (Supplementary Fig. 9a), BiP co-factors that can induce protein degradation[36,43–46]. Of note, a similar absence of such co-chaperone sites has been described for the antibody heavy chain $C_H1$ domain, which is permanently unfolded and only gains structure upon antibody heavy chain-light chain dimerization[17,36,42]. However, since antibody heavy chains are multidomain proteins, chaperone recognition sites can be spatially separated from domains that are well-folded and allow

protein assembly. Such a separation is not possible for the single domain protein IL-23α, where local incomplete folding instead is used for chaperone recognition while preserving assembly-competency. Of note, our HDX measurements reveal helix 4, where a large interaction surface with IL-12β is located[28], to be among the least flexible structural elements in unpaired IL-23α. This may explain how IL-23α can combine assembly-competency with chaperone recognition in another region of the protein, involving its first helix.

Our results show that upon interaction with IL-12β conformational changes occur in IL-23α, prominently involving the first helix but also other parts of the protein, that subsequently prevent chaperone binding and retention. A mutant optimized in silico, IL-23α[opt] stabilized in helix 1, gains structure independently of IL-12β but is still able to form a functional heterodimeric IL-23 complex. These findings suggest that incomplete folding of IL-23α has evolved for quality control and/or regulatory purposes and not for assembly per se. One possible explanation for such a behavior is the combinatorial complexity

of the IL-12 family. Five subunits are used to build at least four different heterodimers, including extensive subunit sharing[47,48]. IL-12β is also part of heterodimeric IL-12, which itself is composed of IL-12α and IL-12β and produced by the same cells as IL-23[49]. ER quality control for IL-23 thus has to monitor the assembly status of IL-23α and at the same time allow for regulation of IL-23 versus IL-12 pairing, which share the same β-subunit. Thus, different quality control mechanisms may exist in immune cells helping to discriminate and regulate IL-12 and IL-23 formation to direct immune responses. Indeed, IL-12α has no free cysteines (besides the IL-12β-interacting cysteine residue), whereas IL-23α additionally possesses two free cysteines in its first helix that strongly participate in its maturation, serving as chaperone anchors for the PDI family member ERp44. Interestingly, ERp44 is regulated by the lower pH in the ERGIC/Golgi compartment[41] but also by zinc ions[50]. Zinc plays pivotal roles in regulating the immune system[51]. Furthermore, it has been shown that zinc upregulates IL-23α mRNA expression[52]. Thus, zinc may not only affect IL-23α on a transcriptional level but could potentially also influence its maturation. Further exploring quality control and assembly mechanisms of IL-12 family members in primary immune cells may thus provide valuable insights into the role of these events in inflammation and immunity.

Engineered cytokines are a powerful tool to modulate immune functions, as previously reported e.g. for IL-2, IL-15, IL-27, and others[53–56]. Directly engineering folding and quality control of interleukins provides one possible avenue to obtain immune signaling molecules not present in nature, but with a low risk for off-target effects or immunogenicity[56]. When assessing the functionality of IL-23α^opt developed in this study, we detected no significant inhibition of IL-23 signaling by isolated IL-23α subunits (Supplementary Fig. 9b, c). Instead, unpaired IL-23α subunits could weakly induce IL-23 signaling in our simplified reporter system (Supplementary Fig. 9d, e), similar to what has been observed for murine and human IL-27α subunits[56,57]. This indicates that unpaired IL-23α is not able to engage and block its receptor and argues for a pronounced participation of IL-12β in initial receptor binding and/or IL-12β-induced structural changes in IL-23α that allow for receptor binding. Furthermore, since IL-23α subunits could weakly induce IL-23 signaling, this possibly indicates even more options for IL-23 receptor activation than a recent study has revealed for the IL-23 heterodimer[58].

Taken together, our study provides detailed structural insights into how protein assembly can be efficiently regulated and controlled in the ER even for single domain proteins: chaperone recognition motifs can be localized in small structural areas, which are sequentially controlled in the secretory pathway and at the same time avoid motifs that would induce premature degradation. Using only small regions for control is compatible with the presence of defined interaction surfaces in the same protein. Molecular insights into these processes can be used to engineer proteins with altered quality control characteristics and desired biological activities.

## Methods
**Constructs**. Human interleukin cDNAs (Origene) were cloned into the pSVL (Amersham) or pcDNA 3.4 TOPO (Gibco) vectors for mammalian expression or the pET21a expression vector (Novagen) for protein production in *E. coli*. Where indicated, proteins contained C-terminal epitope tags separated by a (GS)₅ or (GS)₂ linker. For mammalian expression hamster codon-optimized human interleukin cDNAs (Geneart) were used. The pMT-hamster BiP expression vector[59,60] was a kind gift of Linda Hendershot. Mutants were generated by site-directed mutagenesis. All constructs were sequenced. Sequences of primers used in thus study are given in Supplementary Table 1, sequences of codon-optimized constructs are given in Supplementary Table 2.

**Sequence and structural modeling and analysis**. Multiple sequence alignments were performed using Clustal Omega[61]. Structural alignments were generated with PyMOL (www.pymol.org) based on crystal structures from the PDB database (1F45

(IL-12)[62], 3DUH (IL-23)[28]). Missing loops were modelled with Yasara structure (www.yasara.org) with a subsequent steepest descent energy minimization. Structures were depicted with PyMOL.

**Cell culture and transient transfections**. HEK293T cells were grown in Dulbecco's modified Eagle's medium (DMEM) containing L-Ala-L-Gln (AQmedia, Sigma-Aldrich) supplemented with 10% (v/v) fetal bovine serum (Biochrom or Gibco) at 37 °C and 5% CO₂. Medium was complemented with a 1% (v/v) antibiotic-antimycotic solution (25 µg/ml amphotenicin B, 10 mg/ml streptomycin, and 10,000 units of penicillin; Sigma-Aldrich). Transient transfections were carried out for 24 h either in p35 or p60 poly D-lysine coated dishes (VWR) using GeneCellin (BioCellChallenge) according to the manufacturer's protocol. IL-23α DNA and IL-12β DNA or empty vector (in absence of IL-12β) were (co-)transfected in a ratio of 1:2 for redox-, secretion- and degradation-experiments. Three micrograms IL-23α DNA were used for co-immunoprecipitation experiments. To analyze BiP-interactions, 1 µg hamster BiP DNA was co-transfected with IL-23α.

**Immunoblotting experiments**. For secretion, redox status experiments and knock down experiments with siRNA, cells were transfected for 8 h in p35 dishes, washed twice with PBS and then supplemented with 0.5 ml fresh medium for another 16 h. For siRNA experiments cells were transfected with 25 nM siRNA (Thermo Fisher) for 24 h prior to DNA transfection. siRNA was diluted in Opti-MEM™ Reduced Serum Medium and transfected with Lipofectamine® RNAiMAX Transfection Reagent (Thermo Fisher). For CHX chase assays cells were treated with 50 µg/ml CHX (Sigma-Aldrich) for times indicated in the figures before lysis. Protein half-lives (±SD) were calculated from exponential fits of the curves. To analyze secreted proteins, the medium was centrifuged for 5 min, 300 g, 4 °C. Subsequently, samples were supplemented with 0.1 volumes of 500 mM Tris/HCl, pH 7.5, 1.5 M NaCl (and 200 mM NEM in the case of non-reducing SDS-PAGE) and protease inhibitor and centrifuged for 15 min, 20,000 g, 4 °C. Prior to lysis, if indicated, cells were treated with 10 mM DTT (Sigma-Aldrich) for the last hour or 1 µg/ml Brefeldin A (Sigma-Aldrich) for 2.5 h, washed twice in ice cold PBS, supplemented with 20 mM NEM if samples were to be analyzed by non-reducing SDS-PAGE. Cell lysis was carried out in RIPA buffer (50 mM Tris/HCl, pH 7.5, 150 mM NaCl, 1.0% Nonidet P40 substitute, 0.5% sodium deoxycholate, 0.1% SDS, 1x Roche complete Protease Inhibitor w/o EDTA; Roche Diagnostics) or Triton lysis buffer in the case of co-immunoprecipitation experiments (50 mM Tris/HCl, pH 7.4, 150 mM NaCl, 1 mM EDTA, 1% Triton X-100, 1x Roche complete protease inhibitor w/o EDTA, supplemented with 10 U/ml Apyrase for BiP interaction studies (Sigma-Aldrich) or 20 mM NEM (Sigma) for PDI and Erp44 co-IPs). Samples were supplemented with 0.2 volumes of 5x Laemmli containing either β-Me for reducing SDS-PAGE or 100 mM NEM for non-reducing SDS-PAGE. Deglycosylation assays with Endo H (New England Biolabs), PNGase F (SERVA) or a mix of O-glycosidase and α2–3,6,8 Neuraminidase (New England Biolabs, cleavage of O-glycosylations) were performed according to the manufacturers' protocols. Immunoprecipitations were performed with antibodies against the FLAG-tag, coupled to agarose beads (Sigma-Aldrich, A2220). For each cell lysate of one p60 dish, 30 µl of FLAG beads were used and rotated for 2 h at 4 °C. Immunoprecipitated proteins were washed three times with Triton buffer and eluted with Laemmli buffer for 5 min at 95 °C. For immunoblots, samples were run on 12% SDS-PAGE gels, transferred to PVDF membranes and blotted with anti-myc (Merk Millipore 05–724, 1:1,000 in TBS, 0.05% Tween, 5% milk), anti-FLAG (Sigma-Aldrich F7425, 1:1,000 in TBS, 0.05% Tween, 5% milk), anti-ERp44 (B68)[39] (1:1,000 in TBS, 0.05% Tween, 5% milk), anti-hamster BiP[60] (1:1,000 in TBS, 0.05% Tween, 5% milk), anti-IL-12β (abcam ab133752, 1:500 in TBS, 0.05% Tween, 5% milk), anti-PDI (abcam ab2792, 1:1,000 in TBS, 0.05% Tween, 5% milk) or anti-Hsc70 (Santa Cruz sc-1059, 1:1,000 in gelatin buffer (0.1% gelatin, 15 mM Tris/HCl, pH 7.5, 130 mM NaCl, 1 mM EDTA, 0.1% Triton X-100, 0.002% NaN₃). Species-specific HRP-conjugated secondary antibodies (in TBS, 0.05% Tween, 5% milk or gelatin buffer) were used to detect the proteins (Santa Cruz). Blots were detected using Amersham ECL prime (GE Healthcare) and a Fusion Pulse 6 imager (Vilber Lourmat).

**Immunofluorescence**. A total cell number of $1.2 × 10^4$ COS-7 cells were seeded in µ-slides VI^{0.4} (Ibidi) after transfection with 3.6 µg of DNA using Torpedo transfection reagent (Ibidi). According to manufacturer's instructions, medium was replaced 3 h after seeding. 24 h after transfection, cells were washed twice with PBS at room temperature (RT), and fixed with glyoxal[63] (Sigma-Aldrich). Fixation was performed first for 30 min on ice, then for 30 min at RT. Subsequently samples were quenched for 20 min with NH₄Cl (Sigma-Aldrich). In the following, permeabilization and blocking were performed for 15 min with PBS containing 2.5% BSA and 0.1% Triton-X 100. Primary antibody incubations using mouse monoclonal (M2) anti-FLAG (Sigma-Aldrich, F1804) at 1:500, anti-LMAN1/ERGIC53 (abcam, ab125006) at 1:10 diluted in 2.5% BSA and 0.1% Triton X-100 in PBS or directly purified anti-ERp44 serum provided by Tiziana Anelli (UniSR, Milano) were carried out for 2 h at RT. Following incubation, samples were washed three times with PBS. Handling and usage of fluorophore-conjugated antibodies was performed in the dark. Secondary antibody incubations using anti-mouse-Alexa647 (Cell signaling, #4410 S) at 1:1000 and anti-rabbit-Alexa568 (Thermo Fisher, #A-11011) at 1:1500 diluted in 2.5% BSA and 0.1% Triton X-100 in PBS were carried

out for 1 h and samples subsequently washed with PBS once. Nuclei were stained for 2 min with 0.1 µg/ml DAPI (Sigma-Aldrich). Ultimately, samples were washed another three times with PBS before they were mounted with mounting medium (Ibidi) and imaged. Imaging was performed on Leica DMi8 CS Bino inverted widefield microscope using a $100 \times$ (NA = 1.4) oil immersion objective, and TXR (excitation/bandpass: 560/40 nm; emission/bandpass: 630/75 nm), Y5 (excitation/bandpass: 620/660 nm; emission/bandpass: 700/75 nm), or DAPI (excitation/bandpass: 350/50 nm; emission/bandpass: 460/50 nm) dichroic filters. The LAS X (Leica) analysis software and ImageJ (NIH) where used for image analysis and processing, where adjustments were restricted to homogenous changes in brightness and contrast over each entire image.

**Protein production and purification.** Protein expression in *E. coli* BL21 Star (DE3) (New England Biolabs) of isolated IL-23α subunits (N-terminal His-tag with TEV cleavage site, separated by a 2x GS linker) was performed for 4 h at 37 °C, and resulted in inclusion bodies. Inclusion bodies were solubilized in 50 mM sodium phosphate (pH 7.5), 250 mM NaCl, 10 mM ß-mercaptoethanol, and 6 M guanidine hydrochloride. Solubilized inclusion bodies were centrifuged (20,000 g, 30 min, 20 °C). The supernatant was applied to a HisTrap HP column (GE Healthcare) in 50 mM sodium phosphate (pH 7.5), 250 mM NaCl, 1 mM ß-mercaptoethanol, 6 M guanidine hydrochloride supplemented with 20 mM imidazole. Elution was performed in 50 mM sodium phosphate (pH 4), 250 mM NaCl, 1 mM ß-mercaptoethanol, and 6 M guanidine hydrochloride. Subsequently, the eluate was supplemented with 100 mM DTT and 10 mM EDTA before applying to a HiPrep Sephacryl S-400 HR column (GE Healthcare) equilibrated in 50 mM sodium phosphate (pH 7.5), 6 M guanidine hydrochloride, 1 mM EDTA and 1 mM DTT. After gel filtration, the protein was diluted to 0.1 mg/ml in the same buffer, and then refolded overnight at 4 °C via dialysis against 250 mM Tris/HCl (pH 8.0), 500 mM L-arginine, 100 mM NaCl, 10 mM EDTA, 0.25 mM GSSG, and 0.25 mM GSH. IL-23α$^{VVS}$ was pre-treated with 20 mM GSSG for at least 4 h before refolding. Directly after refolding, the redox status of the purified proteins was assessed by non-reducing SDS-PAGE in the presence of 20 mM NEM. After refolding, IL-23α proteins were dialyzed against 50 mM Tris/HCl (pH 8.0) and the His-tag was cleaved by addition of TEV protease (TEV:IL-23α 1:10 (w/w)) overnight at 4 °C. After cleavage, the protein was applied to a HisTrap HP column (GE Healthcare) in 50 mM Tris/HCl (pH 8.0). The flowthrough was concentrated and applied to an EnrichSEC 70 gel filtration column (BioRad) equilibrated in 10 mM potassium phosphate buffer (pH 7.5). For NMR experiments proteins were purified following the same protocol except that protein producing bacteria were cultured in minimal M9 media supplemented with 1 g/l $^{15}$N-ammonium chloride. IL-23 complexes were produced using the ExpiCHO Expression System (Gibco) according to the manufacturer's protocol. α-subunits (C-terminal His-tag with TEV cleavage site, separated by a GS linker) and ß-subunits (untagged) expressed from the pcDNA3.4 TOPO vector were co-transfected in a ratio of 2:1 for 5 days at 32 °C. After expression, medium was centrifuged (5,000 g, 30 min, 4 °C) and applied to a HisTrap HP column (GE Healthcare) in 10 mM potassium phosphate (pH 7.5), elution was performed in the same buffer with 500 mM imidazole. Subsequently the His-tag was cleaved by addition of TEV protease (TEV:IL-23 complex 1:10 (w/w)) overnight at 4 °C and removed by a second HisTrap HP column in 10 mM potassium phosphate buffer (pH 7.5). Final purification was performed using a HiLoad 26/60 Superdex 200 pg column (GE Healthcare) in 10 mM potassium phosphate (pH 7.5). IL-12β was purified from transiently transfected ExpiCHO (Gibco) cell supernatant according to the manufacturer's. protocol (max. titer). In total, 1 µg IL-12β in the pcDNA3.4 TOPO (C-terminal His-tag with TEV cleavage site, separated by a GS linker) vector was transfected per 1 ml culture. The supernatant was purified using a HisTrap HP column (GE Healthcare) in 10 mM potassium phosphate (pH 7.5) and eluted in the same buffer with 500 mM imidazole. The His-tag was cleaved by addition of TEV protease (TEV:IL-12β, 1:10 (w/w)) overnight at 4 °C in the same buffer. After cleavage, the protein was applied to a HisTrap HP column (GE Healthcare) in the same buffer and the concentrated flowthrough was applied to an EnrichSEC 70 gel filtration column (BioRad) equilibrated in 10 mM potassium phosphate (pH 7.5).

**Protein optical spectroscopy.** Far-UV CD spectra were recorded with a Chirascan plus spectropolarimeter (Applied Photophysics) at 25 °C in 10 mM potassium phosphate, pH 7.5, in 0.2 mm quartz cuvette at a protein concentration of 50 µM. Spectra were recorded 10 times, averaged, and buffer-corrected. Far-UV CD temperature transitions were recorded in a 1 mm quartz cuvette at 10 µM protein concentration in 10 mM potassium phosphate, pH 7.5 at 222 nm. A heating rate of 60 °C/h was used from 25 °C to 95 °C. Near-UV CD temperature transitions were measured in a 2 mm quartz cuvette at 46 µM protein concentration in 10 mM potassium phosphate, pH 7.5 at 260 nm. A heating rate of 60 °C/h was used from 25 °C to 85 °C.

**HDX measurements.** Hydrogen/deuterium exchange (HDX) experiments were performed using an ACQUITY UPLC M-class system equipped with automated HDX technology (Waters, Milford, MA, USA). HDX kinetics for biological duplicates were determined tacking data points at 0, 10, 60, 600, 1800 and 7200 s in

technical triplicates at 20 °C. At each data point of the kinetics, 3 µl of a solution of 30 µM protein were diluted automatically 1:20 into 99.9% D$_2$O-containing 10 mM potassium phosphate, pH 7.5 (titrated with HCl) or the respective H$_2$O–containing reference buffer. The reaction mixture was quenched by the addition of 1:1 200 mM KH$_2$PO$_4$, 200 mM Na$_2$HPO$_4$, pH 2.3 (titrated with HCl), containing 4 M guanidine hydrochloride and 200 mM TCEP at 1 °C and 50 µl of the resulting sample were subjected to on-column peptic digest on a Waters Enzymate BEH pepsin column 2.1 × 30 mm at 20 °C. Peptides were separated by reverse phase chromatography at 0 °C using a Waters Acquity UPLC C18 1.7 µm Vangard 2.1 × 5 mm trapping-column and a Waters Aquity UPLC BEH C18 1.7 µm 1 × 100 mm separation column. For separation a gradient increasing the acetonitrile concentration stepwise from 5–35% in 6 min, from 35–40% in 1 min and from 40–95% in 1 min was applied and the eluted peptides were analyzed using an in-line Synapt G2-S QTOF HDMS mass spectrometer (Waters, Milford, MA, USA). UPLC was performed in protonated solvents (0.1% formic acid), allowing deuterium to be replaced with hydrogen from side chains and amino/carboxyl termini that exchange much faster than backbone amide linkages[64]. All experiments were performed in duplicate. Deuterium levels were not corrected for back exchange and are therefore reported as relative deuterium levels[65]. The use of an automated system handling all samples at identical conditions avoids the need for back exchange correction. MS data were collected over an m/z range of 100–2000. Mass accuracy was secured by calibration with Glu-fibrino peptide B (Waters, Milford, MA, USA) and peptides were identified by MS$^E$ ramping the collision energy automatically from 20–50 V. Data were analyzed in PLGS 3.0.3 and DynamX 3.0 software packages (Waters, Milford, MA, USA).

**Analytical ultracentrifugation.** Sedimentation velocity analytical ultracentrifugation experiments were performed on a ProteomeLab XL-I analytical ultracentrifuge (Beckman Coulter, Brea, CA, USA) equipped with absorbance optics. 350 µl of 10 µM IL-23α in 10 mM potassium phosphate buffer, pH 7.5 were loaded into a standard 12 mm double-sector epon-filled centerpiece, covered with quartz windows, alongside with 420 µl of the reference buffer solution. Samples were centrifuged at 34,000 rpm for IL-23α$^{VVS}$ and 42,000 rpm for IL-23α$^{opt, C54S}$ using an An-50 Ti rotor at 20 °C. Radial absorbance scans were acquired continuously at 230 nm for IL-23α$^{VVS}$ and 235 nm for IL-23α$^{opt, C54S}$ with a radial step size of 0.003 cm. The resulting sedimentation velocity profiles were analyzed using the SedFit software by Peter Schuck with a non-model based continuous Svedberg distribution method (c(s)), with time (TI) and radial (RI) invariant noise on[66]. The density (ρ), viscosity (η) and partial specific volume (v̄) of the potassium phosphate buffer used for data analysis was calculated with SEDNTERP[67].

**Partial proteolysis.** Stability against proteolytic digestion was assessed by partial proteolysis using trypsin gold (VWR). Trypsin was added at a concentration of 1:80 (w/w). Aliquots were withdrawn after different time points, and the proteolysis was terminated by the addition of Roche complete protease inhibitor without EDTA (Roche Applied Science), Laemmli buffer and boiling for 5 min at 90 °C. Proteins were separated on 15% SDS-PAGE gels. Gels were quantified using Fiji ImageJ.

**IL-23α optimization.** IL-23α was optimized using RosettaRemodel to improve stability. The structure of IL-23α was extracted from the chain B of PDB file 5MJ3. IL-23α monomer was first prepared following standard protocols (specified in the flag_relax file) to conform to the Rosetta forcefield. The HDX/NMR data suggested a flexible helix 1, and thus to stabilize the helical bundle, we focused on remodeling the first helix. We first rebuilt the entire helix while allowing the sequence to vary. The first iteration of redocking the helix while redesigning the core is specified in the blueprint and flags file provided (remodel_1.bp and remodel_flags) to stabilize the helix bundle core residues on the first alpha helix, as well as to introduce a helix capping residue (Supplementary Fig. 6a). The top structure from 1000 independent trajectories from the first iteration was chosen based on improved helix core packing and minimal drifting of the first alpha helix. This resulted in mutations Q10A, C14L, L17I, S18I, L21I, and C22L. Leucine on residue 22 impacts the interface with IL-12β, so it was kept as cysteine in the final design, also to preserve one potential ERp44 interaction site. Since Pro9 was unsupported in the IL-23α structure, we extended the N-terminus of the crystal structure by 2 residues, and completely rebuilt the first 6 amino acids in order to create a stable terminus. We incorporated N-capping motifs in residues 7 and 8, as Ser-Pro or Asp-Pro, and tested two different options for residue 6, either as a hydrophobic residue or as part of a salt-bridge with residue 10. This second iteration was run on the aforementioned top structure using remodel_2.bp and the same remodel_flags file but without the -bypass_fragments true flag. 1000 independent trajectories were sampled. After the completion of the two design steps, we cross-referenced by aligning the final design candidates to the crystal structure containing IL-12β and reverted cysteine 22 because the predicted leucine residue would potentially clash with a residue on IL-12β. All residue numbers refer to the IL-23α sequence without the signal peptide.

**NMR spectroscopy.** NMR experiments were performed using $^{15}$N-labeled samples at a concentration of 100 µM in 10 mM KPi (pH 7.5) buffer containing 5% D2O.

¹H, ¹⁵N heteronuclear single quantum coherence (HSQC) experiments using watergate water flip-back for solvent suppression were performed at 298 K on Bruker Avance III spectrometers operating at 900 and 950 MHz proton Larmor frequency using cryogenically cooled probes. Spectra comprised 2046 × 256 complex data points for direct and indirect dimensions, respectively, with a spectral window of 33 ppm centered at 117 ppm for the indirect dimension. All spectra were processed using Bruker Topspin 3.5 software (Bruker, Billerica, USA) using linear prediction and zero filling in the indirect dimension, and were analyzed using CcpNmr Analysis[68].

**IL-23 receptor activation assay**. The IL-23 receptor activation assay was performed using IL-23 iLite® reporter cells from Euro Diagnostica (BM4023), according to the supplier's instructions. To assess activity of recombinant IL-23 complexes a final concentration of 25 ng/ml was used. A final concentration of 25 ng/ml of recombinant human IL-23 (R&D systems, 1290-IL) was used to test IL-23α inhibition. The Firefly and renilla luciferase signal was detected via the Dual-Glo Luciferase Assay System (Promega) in a Spark multimode microplate reader (Tecan group). Individual signals were normalized against the housekeeping renilla signal. To allow better comparisons between mutants, receptor activation of wild type IL-23 was set to 100%.

**Quantification and statistics**. Immunoblots were quantified using the Bio-1D software (Vilber Lourmat). Statistical analyses were performed using Prism (GraphPad Software). Where indicated, data were analyzed with two-tailed, unpaired Student's $t$-tests. Differences were considered statistically significant when $p < 0.05$. Where no statistical data are shown, all experiments were performed at least three times, and one representative experiment was selected.

**Reporting summary**. Further information on research design is available in the Nature Research Reporting Summary linked to this article.

## Data availability

All uncropped immunoblots and SDS-PAGE gels as well as raw data for Fig. 2a and Supplementary Figs. 2a, b, 3c, and 9b–e are provided as a Source Data File. Other data are available from the corresponding author upon reasonable request.

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

## Acknowledgements

We thank Anna Miesl, TUM, for excellent experimental support of the project. We are grateful to Christopher Scheidler and Sabine Schneider, TUM, for access to and help with luminescence measurements. The BiP expression plasmid and antiserum were kind gifts from Linda Hendershot, St. Jude Children's Research Hospital, Memphis/TN, USA. The ERp44 antibodies were kind gifts from Tiziana Anelli and Roberto Sitia, Ospedale San Raffaele, Milan/Italy. We acknowledge access to NMR measurements at the Bavarian NMR Center. JEvB gratefully acknowledges funding by a Helmholtz Young Investigator grant (VH-NG-1331). MJF is a Rudolf Mößbauer Tenure Track Professor and as such gratefully acknowledges funding through the Marie Curie COFUND program and the Technische Universität München Institute for Advanced Study, funded by the German Excellence Initiative and the European Union Seventh Framework Program under Grant Agreement 291763. This work was performed in the framework of SFB 1035 (German Research Foundation DFG, Sonderforschungsbereich 1035, projects A03, B11 and Z1).

## Author contributions

M.J.F. conceived the study. N.M.R. experiments were performed by A.L. and analyzed by A.L. and M.S. P.W.N.S. performed ultracentrifugation experiments, N.B. performed microscopy experiments. HDX experiments were carried out and analyzed by S.B., F.R., and M.H. All other experiments were performed by S.M., S.B., and C.J.K. In silico protein optimizations were performed by C.A.C. and P.S.H. Data were analyzed by all authors and the paper was written by S.M. and M.J.F. with input from S.M., S.B., C.J.K., A.L., M.H., M.S., J.E.v.B., and P.S.H.

## Additional information

**Competing interests:** The authors declare no competing interests.

