## [Peer Review File · Nature Communications]

Reviewers' comments:

Reviewer #1 (Remarks to the Author):

The authors address an interesting conundrum regarding the quality control of luminal and secreted protein complexes: how do chaperones and degradation machinery distinguish misfolded proteins from complex subunits awaiting a binding part? This problem is compounded by the necessity to match appropriate cysteine pairs for disulfide bond formation. To test quality control mechanisms of secreted protein complexes, the authors selected the IL-23 cytokine as an example heterodimer. The study integrates functional secretion data and biophysical approaches to probe protein dynamics and complex formation. The results highlight the importance of a semi-stable helix in the IL-23a subunit as a marker for heterodimer assembly status. Apparently, the IL-23a subunit can sufficiently fold to dodge ER folding quality control. However, the monomeric IL-23a retains high flexibility/dynamics in helix 1, a structural motif that also exhibits an unusual pair of free, reduced cysteines. According to the proposed model, the exposure of the free helix 1 cysteines attracts the protein disulfide isomerase ERp44 and ubiquitous ER chaperone BiP. Satisfactory heterodimer formation would potentially release these chaperones for secretion of mature IL-23.

The founding question of this study is very interesting, and the proposed mechanism may have relevance to many other secreted protein complexes. The study probes both structural details and cellular trafficking. Structural insights are gleaned from biophysical techniques sensitive to protein dynamics (e.g., NMR and HDX). These approaches yielded unique insights into the folding stability of the IL-23 subunit at the cusp of complex formation. Furthermore, the authors test their model by re-engineering the "marker" helix 1 to improve its stability as a monomer. As predicted by the proposed model, the stabilized marker helix allowed for premature secretion of the monomer.

This work is broadly interesting to the protein quality control community, addressing a unique niche in the problems of ER protein assembly. The HDX methodology is well-documented, and HDX results are interpreted with appropriate restraint/context. However, the results could be better served by reporting more HDX data (see below). I have a few points that may improve the reporting and quality of the research communication below.

Major Points:

1. To enable a biophysical characterization of the folding and heterodimerization of IL-23a, the authors generate a triple mutant [IL-23a(VVS)] with the pre-assembly free thiols removed. Considerable effort is made to characterize the stability of this variant. However, it is difficult to assess the impact of the triple mutant on the native stability of the subunit. Can the authors compare the stability of this IL-23a(VVS) with the native IL-23a(WT) using AUC, Far-UV CD, and/or HSQC NMR? The comparison to WT seems like an important baseline for comparison, unless there are extenuating circumstances. Do the extra three unpaired cysteines make IL-23a(WT) difficult to purify?
2. It is not always clear which experiments were performed with IL-23a(WT) vs. IL-23a(VVS) variant, especially in the HDX experiments. If I understand correctly, "IL-23a in isolation" (pg. 9 line 213) refers to IL-23a(VVS), based on the Fig. 3 legend. This should be clarified in the main text. Is the IL-23a/IL-12b complex also using the VVS variant here? If so, the authors should address the implications of measuring a non-disulfide bonded version of the heterodimer here. For example, will this association be strong enough to favor a homogenous population of 1:1 IL-23a/IL-12b complexes without the disulfide bond? Can the K_d be estimated? These issues are not likely to change the overall interpretation, but the clarifications can help in evaluating the magnitude of the HDX perturbations.
3. Fig. 4a appears to use a reflected image of the IL-23a structure. This is an inappropriate manipulation of the structure. Furthermore, it's not clear what the reflected/"optimized H1" structure adds to the figure. The "a" panel could be reduced to the IL-23a structure (with unstable H1 highlighted), accompanied by the sequences as currently annotated.

4. The HDX data is nicely presented in Figs 3 & 4, and the methods are largely well-documented. However, the data are highly processed, and there are no data figures presented to assess the quality and significance of the reported changes in dynamics. Admittedly, there is no universal standard for reporting HDX data, but the authors could generate a few more figures to clarify the HDX data without much extra effort. For example, an SI figure showing peptic peptide coverage maps would communicate sequence coverage and redundancy of the measurements. Furthermore, significance testing was not reported for differences in fractional uptake between treatments. Again, this testing is not universally applied in HDX data, but some example uptake plots could clarify the magnitude of the uptake rate differences and the reproducibility of the technical replicates. Of course, it would be perfectly reasonable to relegate these plots to the SI section.

Minor Points:

Fig. 3 and Supp. Fig. 3: What is meant by "Helix 4.. is not fully resolved"? Non-resolved by HDX-MS? Or unresolved in the structural model to which the HDX data is mapped? I presume it is the former circumstance, and as indicated in Major point #4, a HDX coverage map would resolve this ambiguity.

SI Fig. 3h. Can the authors clarify the cytokine concentrations used in the IL-23 receptor activation assay in the figure legend?

The abstract could use some proof-reading. For example, typo line 23; awkward phrasing in line 30; and "abundant protein topology" (line 33) is ambiguous.

Reviewer #2 (Remarks to the Author):

Meier et al. Nat Comm.

In this interesting study, Meier et al use heterodimeric interleukins to tackle a fundamental problem in molecular cell biology, how cells couple folding and assembly of their secretory products. The senior author of the manuscript pioneered this field showing that the folding of the CH1 domain of immunoglobulin heavy chains is promoted by light chain assembly. Here the attention focuses on single domain proteins, namely interleukins 6, 12 & 23.

The results show that partial unfolding in the first helix of IL23 alpha chains induce sequential binding to BiP and ERp44. Excitingly, these interactions prevent degradation of unassembled alpha chains. The authors propose that this sophisticated quality control mechanisms allow the regulated assembly of different interleukins sharing the same beta chain.

Once identified the main underlying features, the authors generate mutant interleukin 23 alpha chains that can be secreted without beta chains and maintains some of the activity of the original heterodimer.

In general, the results are convincing. There are however a number of points that the authors should consider attentively before the manuscripts goes into print.

Considering that ERp44 binding to C14 or C22 prevents the degradation of unassembled alpha chains, cells lacking ERp44 might secrete less IL23, and/or shift to the production of IL12. This prediction should be easy to test. Likewise, manipulating pH or zinc concentration in the secretory pathway to inhibit ERp44 activity might yield interesting phenotypes.

Figure 1C

In non-reducing conditions, most IL23 alpha yields several bands of 40-50 kDa or more. What are they? Which of the three free cysteines are involved in their formation?

Figure 1f, and line 118

Given their different cysteine arrangement, do IL6 and IL12 behave differently with respect to

stability and ERp44 binding?

Figure 1g

Almost no IL23 alpha C58-70 is detectable in the lysates, unless beta is coexpressed. Is this because of rapid degradation (despite the C14-22 mutant is even more unstable)? Do proteasome or lysosomal inhibitors stabilize it?

Figure 2b

Left panel: Any cue as to why the 58-70 mutant runs faster?

Center panel: This point is particularly critical. If the same data were to be plotted with the same scale (0-250%), it would seem that the mutants C54S and C14-22V bind much less BiP, even if a ns label is given to them. This is even more evident looking at supplementary Figure S2f: the 14-22 double mutant binds much lower amounts of both ERp44 and BiP.

This would point at the evolution of colocalized determinants for BiP and ERp44. Could ERp44 compete with BiP binding, thereby retarding degradation.

It might be worth to see if the mutants differ in the mechanisms of degradation, besides in their half-lives. It also seems that there is a bimodal distribution of the single experiments, only 2 or 3 of them yielding more BiP binding. What is the possible reason for these remarkable differences?

Figure 4b

Is the site of O glycosylation known?

Supplementary Figure S5b-c

It would be better to use the same scale, to highlight that the engineered opt variant is more active than VVS.

Graphical abstract

The scheme may be interpreted to suggest that BiP acted after ERp44 in the biogenesis of IL23. This is possible, though in view of the colocalisation of IL23alpha and ERp44 in ERGIC, the reverse is more likely. In this connection, I'd suggest to swap the order of the plots in Figure 2b, center panel. Once put in the same scale, BiP should be at the first of ERp44, assuming that this is the order of events. Indeed inhibiting the formation of the intrachain disulfide bonds (ER oxidative folding) induces binding to BiP much more than to ERp44.

Going back to the graphical abstract, the authors should introduce degradation in the image, to highlight the proposal that the first helix evolved as a chaperone binder that retard degradation and favors assembly. Is there evidence that ERp44 binds to C54. Does PDI bind to C54?

Reviewer #3 (Remarks to the Author):

The submitted paper describes an interesting study of the molecular mechanisms used by mammalian cells to ensure the secretion of correctly folded and assembled heterodimeric cytokines such as interleukin-23 (IL-23). The paper is clearly written with well presented and described figures. The majority of the conclusions drawn from the results presented appear to be sound, however, I have some questions concerning the interpretation of NMR and CD data, which are summarised below.

i) The HSQC NMR spectrum shown for the VVS variant of IL-23 α contains features typical of a partially folded protein, with some relatively well dispersed backbone amide signals indicative of a folded region, but many fairly broad signals at close to random coil positions suggesting a molten globule-like state for substantial areas of the protein. The authors report no attempt to obtain sequence-specific backbone resonance assignments for the variant of IL-23 α but have produced single site-directed variants of the four tryptophan residues to assign the indole side chain signals observed. This approach clearly shows that the side chain of tryptophan 26 (W26) exists in two states, which interconvert on a relatively slow timescale. The authors interpret this finding as evidence for two conformational states of helix 1, however, the side chain of W26 appears to be located on the surface of IL-23 α (see figure 3e), so the two states observed for W26 could simply

be two distinct local conformations for the W26 side chain.

ii) The authors also state that the binding of IL-12 β to the VVS variant of IL-23 α results in the major NMR peak for the free form of the W26 side chain losing its intensity to the minor W26 peak observed for the free protein. From the region of the HSQC spectrum shown in figure 3g it appears that the W26 peak corresponding to the major form of the free IL-23 α is significantly reduced on IL-12 β binding, however, this does not seem to be accompanied by a significant increase in the intensity of the minor W26 peak. I think that the the authors may be over interpreting the changes in the W26 side chain as an indicator of conformational changes/dynamics in the whole of helix 1 of IL-23 α .

iii) The authors present CD spectra-based temperature stability data for both the VVS variant of IL-23 α (figure 3b) and a variant designed to have a fully stabile helix 1 (figure 4e). The results shown for the helix-1 optimised variant are interpreted as evidence for a fully folded IL-23 α , however, this is not consistent with an essentially linear loss of helical structure with increasing temperature, rather than the highly cooperative unfolding expected for a fully folded protein. The authors should reconsider the interpretation of this data and its significance for the mechanistic model presented in the paper.

Overall, the molecular mechanism proposed by the authors is largely supported by the interpretation of the results presented, however, the precise conformational state of helix 1 in IL-23 α is probably less clear than currently presented in the paper (perhaps an equilibrium between partially folded and folded states). There is also evidence of conformational instability/dynamics in at least one other helix of IL-23 α from the HDX data shown in figure 3d, which should be considered.

Reviewed by Professor Mark Carr

Dear reviewers,

We would like to thank you very much for the positive evaluation as well as the constructive criticism on our study. Please find below a point-by-point reply where we have addressed your concerns by additional experiments as well as by editing the text and figures of our manuscript. Major text changes in the manuscript are additionally highlighted in one of the submitted files. We hope that our reply addresses all your concerns adequately and would like to thank you again for the time you have invested in reviewing our manuscript.

Reviewer 1:

The authors address an interesting conundrum regarding the quality control of luminal and secreted protein complexes: how do chaperones and degradation machinery distinguish misfolded proteins from complex subunits awaiting a binding part? This problem is compounded by the necessity to match appropriate cysteine pairs for disulfide bond formation. To test quality control mechanisms of secreted protein complexes, the authors selected the IL-23 cytokine as an example heterodimer. The study integrates functional secretion data and biophysical approaches to probe protein dynamics and complex formation. The results highlight the importance of a semi-stable helix in the IL-23a subunit as a marker for heterodimer assembly status. Apparently, the IL-23a subunit can sufficiently fold to dodge ER folding quality control. However, the monomeric IL-23a retains high flexibility/dynamics in helix 1, a structural motif that also exhibits an unusual pair of free, reduced cysteines. According to the proposed model, the exposure of the free helix 1 cysteines attracts the protein disulfide isomerase ERp44 and ubiquitous ER chaperone BiP. Satisfactory heterodimer formation would potentially release these chaperones for secretion of mature IL-23.

The founding question of this study is very interesting, and the proposed mechanism may have relevance to many other secreted protein complexes. The study probes both structural details and cellular trafficking. Structural insights are gleaned from biophysical techniques sensitive to protein dynamics (e.g., NMR and HDX). These approaches yielded unique insights into the folding stability of the IL-23 subunit at the cusp of complex formation. Furthermore, the authors test their model by re-engineering

the “marker” helix 1 to improve its stability as a monomer. As predicted by the proposed model, the stabilized marker helix allowed for premature secretion of the monomer.

This work is broadly interesting to the protein quality control community, addressing a unique niche in the problems of ER protein assembly. The HDX methodology is well-documented, and HDX results are interpreted with appropriate restraint/context. However, the results could be better served by reporting more HDX data (see below). I have a few points that may improve the reporting and quality of the research communication below.

Major Points:

1. To enable a biophysical characterization of the folding and heterodimerization of IL-23a, the authors generate a triple mutant [IL-23a(VVS)] with the pre-assembly free thiols removed. Considerable effort is made to characterize the stability of this variant. However, it is difficult to assess the impact of the triple mutant on the native stability of the subunit. Can the authors compare the stability of this IL-23a(VVS) with the native IL-23a(WT) using AUC, Far-UV CD, and/or HSQC NMR? The comparison to WT seems like an important baseline for comparison, unless there are extenuating circumstances. Do the extra three unpaired cysteines make IL-23a(WT) difficult to purify?

The reviewer raises an important point, i.e.: benchmarking our triple mutant to the IL-23a wt protein. We have tried multiple different conditions to produce and refold IL-23a wt – which unfortunately under all conditions tested covalently misfolds, most likely due to its free cysteines that are recognized and shielded by PDI family members in the secretory pathway. This was the major reason why we chose, and characterized in detail, the triple mutant. We hope that the reviewer agrees with this approach and would like to point out that whenever we could draw comparisons, the triple mutant and the IL-23a wt protein behaved highly similar. Furthermore, our study contains an additional triple mutant as a further control (where all three free Cys are replaced by serines), which also behaved highly similar to IL-23a wt as well as the VVS mutant whenever comparisons were possible. And lastly, we now include new data in SI Figure 3

and 4, which show HDX measurements for the wt IL23 complex. We hope that together this addresses the reviewer's concerns. We have also now more clearly stated the reason for our approach in the revised manuscript.

2. It is not always clear which experiments were performed with IL-23a(WT) vs. IL-23a(VVS) variant, especially in the HDX experiments. If I understand correctly, "IL-23a in isolation" (pg. 9 line 213) refers to IL-23a(VVS), based on the Fig. 3 legend. This should be clarified in the main text. Is the IL-23a/IL-12b complex also using the VVS variant here? If so, the authors should address the implications of measuring a non-disulfide bonded version of the heterodimer here. For example, will this association be strong enough to favor a homogenous population of 1:1 IL-23a/IL-12b complexes without the disulfide bond? Can the Kd be estimated? These issues are not likely to change the overall interpretation, but the clarifications can help in evaluating the magnitude of the HDX perturbations.

We apologize for not having been clear enough in this part. We now clearly state whenever the VVS mutant was used. Furthermore, for all HDX measurements of heterodimers, the interchain disulfide bond was always preserved to allow for quantitative complex formation, which otherwise could lead to misinterpretation of our data as the reviewer correctly points out. This fact is now also more clearly mentioned in the revised manuscript.

3. Fig. 4a appears to use a reflected image of the IL-23a structure. This is an inappropriate manipulation of the structure. Furthermore, it's not clear what the reflected/"optimized H1" structure adds to the figure. The "a" panel could be reduced to the IL-23a structure (with unstable H1 highlighted), accompanied by the sequences as currently annotated.

We have adjusted the Figure as suggested by the reviewer.

4. The HDX data is nicely presented in Figs 3 & 4, and the methods are largely well-documented. However, the data are highly processed, and there are no data figures presented to assess the quality and significance of the reported changes in dynamics. Admittedly, there is no universal standard for reporting HDX data, but the authors could generate a few more figures to clarify the HDX data without much extra effort. For

example, an SI figure showing peptic peptide coverage maps would communicate sequence coverage and redundancy of the measurements. Furthermore, significance testing was not reported for differences in fractional uptake between treatments. Again, this testing is not universally applied in HDX data, but some example uptake plots could clarify the magnitude of the uptake rate differences and the reproducibility of the technical replicates. Of course, it would be perfectly reasonable to relegate these plots to the SI section.

As suggested by the reviewer, we now include in the SI material new figures (SI Figures 4 and 8) that show more unprocessed data of our HDX measurements.

In detail, we have now included:

- Peptide coverage maps for all proteins measured
- Exchange graphs for individual peptides

We hope the reviewer is satisfied with these adjustments.

Minor Points:

Fig. 3 and Supp. Fig. 3: What is meant by “Helix 4.. is not fully resolved”? Non-resolved by HDX-MS? Or unresolved in the structural model to which the HDX data is mapped? I presume it is the former circumstance, and as indicated in Major point #4, a HDX coverage map would resolve this ambiguity.

The reviewer is correct in the assumption that we referred to “not fully resolved in the HDX” measurements. As suggested, to remove this ambiguity, we have now included coverage maps in the SI. Furthermore, in all structural models we now consistently indicate parts not resolved in HDX measurements in the revised figures.

SI Fig. 3h. Can the authors clarify the cytokine concentrations used in the IL-23 receptor activation assay in the figure legend?

As suggested by the reviewer, we have added this information to the figure legend.

The abstract could use some proof-reading. For example, typo line 23; awkward phrasing in line 30; and “abundant protein topology” (line 33) is ambiguous.

We have edited the abstract as suggested by the reviewer.

Reviewer #2:

In this interesting study, Meier et al use heterodimeric interleukins to tackle a fundamental problem in molecular cell biology, how cells couple folding and assembly of their secretory products. The senior author of the manuscript pioneered this field showing that the folding of the CH1 domain of immunoglobulin heavy chains is promoted by light chain assembly. Here the attention focuses on single domain proteins, namely interleukins 6, 12 & 23.

The results show that partial unfolding in the first helix of IL23 alpha chains induce sequential binding to BiP and ERp44. Excitingly, these interactions prevent degradation of unassembled alpha chains. The authors propose that this sophisticated quality control mechanisms allow the regulated assembly of different interleukins sharing the same beta chain.

Once identified the main underlying features, the authors generate mutant interleukin 23 alpha chains that can be secreted without beta chains and maintains some of the activity of the original heterodimer.

In general, the results are convincing. There are however a number of points that the authors should consider attentively before the manuscripts goes into print.

Considering that ERp44 binding to C14 or C22 prevents the degradation of unassembled alpha chains, cells lacking ERp44 might secrete less IL23, and/or shift to the production of IL12. This prediction should be easy to test. Likewise, manipulating pH or zinc concentration in the secretory pathway to inhibit ERp44 activity might yield interesting phenotypes.

This is a very valid comment by the reviewer, which we addressed experimentally: using siRNA we knocked down ERp44 and tested, if under these conditions IL23 α could become secreted in isolation, which would argue for a major role of ERp44 in retaining IL23 α . Indeed, knockdown of ERp44 led to an increase in secretion of unpaired IL23 α , but not complete secretion, which

is consistent with an important but not exclusive role for ERp44 in IL23 α assembly control. This new data has been included in SI Figure 2. Assessing changes in relative IL12/23 secretion is not trivial, but very interesting, and will thus be the focus of future studies.

Figure 1C

In non-reducing conditions, most IL23 alpha yields several bands of 40-50 kDa or more. What are they? Which of the three free cysteines are involved in their formation?

To address this question, we have performed additional experiments. First, we quantified the amount of higher molecular weight species for the different cysteine mutants of IL23 α (SI Figure 2b). Whereas no mutant completely abolished the formation of high MW species, the VVS mutant reduced them most strongly. Second, we assessed if IL23 α could form homodimers, as we have described for IL12 α (Reitberger et al, JBC, 2017). Indeed, IL23 α formed homodimers in these experiments (SI Figure 2c). Taken together, these data argue that most cysteines of IL23 α are (partially) involved in the formation of high MW covalent species and that these species involve also homomeric assemblies.

Figure 1f, and line 118

Given their different cysteine arrangement, do IL6 and IL12 behave differently with respect to stability and ERp44 binding?

IL12a has a longer half-life in cells of approximately 1.5-2h (Reitberger et al, JBC, 2017) compared to the half-life of ca. 60 min we determined for IL23a in this study. In preliminary work (unpublished as of yet) we have assessed binding of ERp44 to IL12 and IL23, not to IL6, which becomes secreted. In these experiments we found ERp44 to bind to IL12 and IL23, which argues that ERp44 performs a more general role in the quality control of IL12 family cytokines. For IL12, however, we could not yet define specific cysteines/sites ERp44 binds to, which we plan to address in a future study.

Figure 1g

Almost no IL23 alpha C58-70 is detectable in the lysates, unless beta is coexpressed. Is this because of rapid degradation (despite the C14-22 mutant is even more unstable)? Do proteasome or lysosomal inhibitors stabilize it?

Based on this comment we have repeated this experiment several more times. We observe heterogeneity in the expression levels of IL23 alpha C58-70S, but not consistently a lower level. We this have replaced the panel with a more representative blot.

Figure 2b

Left panel: Any cue as to why the 58-70 mutant runs faster?

This mutant contains a shorter GS-linker between the protein and the FLAG-tag. We excuse not having mentioned this in the figure legend, which we now do.

Center panel: This point is particularly critical. If the same data were to be plotted with the same scale (0-250%), it would seem that the mutants C54S and C14-22V bind much less BiP, even if a ns label is given to them. This is even more evident looking at supplementary Figure S2f: the 14-22 double mutant binds much lower amounts of both ERp44 and BiP. This would point at the evolution of colocalized determinants for BiP and ERp44. Could ERp44 compete with BiP binding, thereby retarding degradation. It might be worth to see if the mutants differ in the mechanisms of degradation, besides in their half-lives. It also seems that there is a bimodal distribution of the single experiments, only 2 or 3 of them yielding more BiP binding. What is the possible reason for these remarkable differences?

Based on this very good observation of the reviewer, we first adjusted the scales to facilitate comparisons. Second, we significantly extended our co-IP experiments by the addition of more datasets to obtain statistically even more robust data (at least seven, but in several cases now more than ten independent repeats). These data corroborate our findings that no significant difference in terms of BiP binding exists for the C14,22V mutant. However, indeed the lower binding of C54S to BiP is rendered significant by the addition of more datasets (Fig. 2b) and the binding of ERp44 to C22S has become non-significant (SI Fig.

2e). Thus, trends we have observed are now on a more solid statistical basis. The apparent bi-modal distribution is now much less pronounced, and was thus most likely routed in chance. Apart from these changes, we completely agree with the reviewer that a general trend seems to exist: Whenever ERp44 binding is affected BiP seems to be affected (at least qualitatively) similarly. This indeed points towards the evolution of co-localized binding determinants, which is in agreement with the notion that helix 1 is a major binding site for both chaperones. Based on these findings, future studies now indeed should address the exciting question if an either-or binding mode exists and what the consequences for the fate of IL23 α are. These new findings (and their possible interpretation) are now also included in the revised manuscript.

Figure 4b

Is the site of O glycosylation known?

As we agree with the reviewer that this is an important question, which may also provide further structural insights into IL23 assembly, we decided to identify the O-glycosylation site in IL23 α . We used an *in silico* prediction of residues with a probability to become O-glycosylated (NetOGlyc), and among those narrowed our search to surface-exposed ones that would become buried upon IL23 α interaction with IL12 β . Thr 167 was identified as a candidate like this. And indeed, replacing Thr 167 by Gly abolished O-glycosylation of IL23 α , which is fully consistent with IL12 β blocking the O-glycosylation site as previously hypothesized by us. This new data is now shown in SI Figure 6c.

Supplementary Figure S5b-c

It would be better to use the same scale, to highlight that the engineered opt variant is more active than VVS.

As suggested by the reviewer, we have adjusted the scales.

Graphical abstract

The scheme may be interpreted to suggest that BiP acted after ERp44 in the biogenesis of IL23. This is possible, though in view of the colocalisation of IL23alpha and ERp44 in ERGIC, the reverse is more likely. In this connection, I'd suggest to swap

the order of the plots in Figure 2b, center panel. Once put in the same scale, BiP should be at the first of ERp44, assuming that this is the order of events. Indeed inhibiting the formation of the intrachain disulfide bonds (ER oxidative folding) induces binding to BiP much more than to ERp44.

Going back to the graphical abstract, the authors should introduce degradation in the image, to highlight the proposal that the first helix evolved as a chaperone binder that retard degradation and favors assembly.

As rightfully suggested by the reviewer, ERp44 is more likely to act after BiP. Thus, to make this notion more accessible to the reader we have swapped the BiP/ERp44 binding panels in all main and SI Figures and at several places in the text. Furthermore, in the legend to Figure 5 (the model) we now mention this more likely order of events. Lastly, we now have added degradation into the model.

Is there evidence that ERp44 binds to C54. Does PDI bind to C54?

We now have addressed this question in more detail experimentally and indeed as reported in our initial submission found ERp44 to bind to C54 (Figure 2b), which is consistent with the fact, that we need to replace C14, C22 and C54 to observe secretion of unpaired IL23 α . In terms of PDI binding, we performed additional experiments but did not observe significant differences between different mutants we tested, i.e.: no indication for a preferential binding to C54 (SI Figure 2d).

Reviewer #3 :

The submitted paper describes an interesting study of the molecular mechanisms used by mammalian cells to ensure the secretion of correctly folded and assembled heterodimeric cytokines such as interleukin-23 (IL-23). The paper is clearly written with well presented and described figures. The majority of the conclusions drawn from the results presented appear to be sound, however, I have some questions concerning the interpretation of NMR and CD data, which are summarised below.

i) The HSQC NMR spectrum shown for the VVS variant of IL-23 α contains features typical of a partially folded protein, with some relatively well dispersed backbone amide

signals indicative of a folded region, but many fairly broad signals at close to random coil positions suggesting a molten globule-like state for substantial areas of the protein. The authors report no attempt to obtain sequence-specific backbone resonance assignments for the variant of IL-23 α but have produced single site-directed variants of the four tryptophan residues to assign the indole side chain signals observed. This approach clearly shows that the side chain of tryptophan 26 (W26) exists in two states, which interconvert on a relatively slow timescale. The authors interpret this finding as evidence for two conformational states of helix 1, however, the side chain of W26 appears to be located on the surface of IL-23 α (see figure 3e), so the two states observed for W26 could simply be two distinct local conformations for the W26 side chain.

We agree with the reviewers' impression that the HSQC spectrum is indicative of folded (with linewidths) and (partially) misfolded signals (with significant line-broadening). We have indeed attempted to record HNCA experiments using a ^{13}C , ^{15}N labeled sample. However, limited solubility and aggregation at higher concentrations rendered this experiment too insensitive, thus prohibiting backbone chemical shift assignments. We therefore analyzed the well-resolved Trp signals, which we could assign by mutagenesis.

An estimate counting of the backbone signals with narrower linewidth suggests that there might be 99 out of 131 observed amides corresponding to well-folded regions. The number of signals that experience significant line-broadening can be estimated to be 32. This appears consistent with the number of residues comprising helix 1 and proximal regions experiencing line-broadening due to local unfolding of helix 1. Note that, given the helical conformation of the protein, a relatively narrow ^1H chemical shift dispersion is expected and thus presumably many narrow signals corresponding to folded regions are overlapped with the broadened signals.

We would like to emphasize that several (unassigned) backbone amide signals in the spectra show peak splitting, and collapse into one of the signals upon IL12 β binding is also observed for these peaks, suggesting that our observations for the W26 side chain are due to a conformational exchange

affecting multiple residues and not a single indole ring flip (please see figure below for examples), which is also consistent with our HDX mass spectrometry data.

ii) The authors also state that the binding of IL-12 β to the VVS variant of IL-23 α results in the major NMR peak for the free form of the W26 side chain losing its intensity to the minor W26 peak observed for the free protein. From the region of the HSQC spectrum shown in figure 3g it appears that the W26 peak corresponding to the major form of the free IL-23 α is significantly reduced on IL-12 β binding, however, this does not seem to be accompanied by a significant increase in the intensity of the minor W26 peak. I think that the authors may be over interpreting the changes in the W26 side chain as an indicator of conformational changes/dynamics in the whole of helix 1 of IL-23 α .

We believe that the reduction of the major peak of the W26 side chain, and also of the backbone amide signals, which exhibit peak splittings (see above) demonstrates that one of the two conformations is affected by IL12 β binding. Please note, that the plot levels of free and IL12 β bound spectra are not directly comparable in Fig. 3g in spite of the two experiments having been performed under the same conditions (i.e. magnetic field, buffer, number of scans and protein concentration). As the more than two-fold increase of molecular weight

in the complex increases the linewidths and leads to a corresponding reduction of signal-to-noise, we have scaled the plot levels in the bound form to be comparable to the free form. We therefore would argue that the relative intensities of the two signals corresponding to the W26 side chain at least qualitatively report on the relative population of free and IL12 β bound forms. To further clarify this, the intensity ratios are calculated as the height ratios between the minor and major peaks of the free form of IL23, and the corresponding signals in the complexed form. A shift towards one of the pre-existing conformations is observed in all cases, although shift does not always happen towards the minor conformation as observed for the W26 side chain. This likely reflects different relaxation properties of side chain vs. backbone residues.

Peak	I(minor)/I(main) free IL23 α	Intensity ratio IL23 α - IL12 β complex
W26 (reference)	0.3	1.9
Peak 1	0.7	0.9
Peak 2	0.7	Total shift
Peak 3	0.8	Total shift
Peak 4	0.7	Total shift

iii) The authors present CD spectra-based temperature stability data for both the VVS variant of IL-23 α (figure 3b) and a variant designed to have a fully stable helix 1 (figure 4e). The results shown for the helix-1 optimised variant are interpreted as evidence for a fully folded IL-23 α , however, this is not consistent with an essentially linear loss of helical structure with increasing temperature, rather than the highly cooperative unfolding expected for a fully folded protein. The authors should reconsider the interpretation of this data and its significance for the mechanistic model presented in the paper.

This indeed is an important point raised by the reviewer. Our *in silico* optimization may have stabilized helix 1 to such an extent, that it unfolds even after the remainder of the protein upon heating, which would give rise to a less-cooperative unfolding transition. We thus decided to additionally measure NUV-CD temperature transitions for the VVS mutant and the mutant with the

optimized helix to additionally measure changes in tertiary structure. Our results reveal i) very similar melting points for both methods in each case ii) again, a cooperative transition for VVS and a non-cooperative transition for the helix-1 optimized variant. These new data are now included in SI Figure 7c and d. Based on these findings we favor an interpretation that whereas optimization of helix leads indeed to the desired stabilization of this structural element (as our cellular and HDX data show), and thus the ability of the protein to bypass ER quality control, but we induce a loss in cooperativity potentially due to loss of important tertiary interactions and/or the introduction of a small autonomously folding-competent folding element (helix 1). We now discuss these findings in more detail and in the revised manuscript reworded our statements about the overall stabilization of IL23a by helix-1 optimization to be more cautious with the interpretation.

Overall, the molecular mechanism proposed by the authors is largely supported by the interpretation of the results presented, however, the precise conformational state of helix 1 in IL-23a is probably less clear than currently presented in the paper (perhaps an equilibrium between partially folded and folded states). There is also evidence of conformational instability/dynamics in at least one other helix of IL-23a from the HDX data shown in figure 3d, which should be considered.

Based on this comment, we have at several places throughout the revised manuscript adjusted our wording to be more cautious in terms of helix 1 being unstructured in comparison to the remainder of the protein. We agree with the reviewer that in a small single domain protein unfolding of one part will most likely influence the folding state of the rest of the protein. Indeed, as also suggested by our NMR data, an equilibrium between different degrees of folding is likely, which we now also state more clearly.

REVIEWERS' COMMENTS:

Reviewer #1 (Remarks to the Author):

I am satisfied with the changes made by the authors in response to my comments. The coverage map is helpful, and the uptake plots demonstrate reproducibility. As for the WT misfolding, this is an understandable hurdle, and I am satisfied that the authors made comparisons between VVS variant and the WT where possible.

Reviewer #3 (Remarks to the Author):

The authors have now convincingly addressed all the queries and concerns identified in the original version of the manuscript, including the inclusion of additional experimental data in the revised paper and in places a more cautious interpretation of the results presented. The paper reports an interesting and important advance in our understanding of the mechanisms underpinning quality control processes for secreted protein complexes, which is based on sound interpretation of a comprehensive and complementary set of experimental data. I strongly recommend acceptance of the revised paper for publication.

Reviewer: Prof. Mark Carr